immunology, ecology, evolution

seasonal infection, photoperiod, seasonal immune responses, stickleback, host–parasite interactions

**Author for correspondence:**
James R. Whiting
e-mail: j.whiting2@exeter.ac.uk

# Prior exposure to long-day photoperiods alters immune responses and increases susceptibility to parasitic infection in stickleback

James R. Whiting[1,2], Muayad A. Mahmud[1,3], Janette E. Bradley[1] and Andrew D. C. MacColl[1]

[1]School of Life Sciences, University of Nottingham, University Park, Nottingham NG7 2RD, UK
[2]Department of Biosciences, University of Exeter, Geoffrey Pope Building, Exeter EX4 4QD, UK
[3]Scientific Research Center, Erbil Polytechnic University, Erbil, Iraq

JRW, 0000-0001-8936-4991; MAM, 0000-0001-8898-6393; JEB, 0000-0003-3973-7977; ADCM, 0000-0003-2102-6130

Seasonal disease and parasitic infection are common across organisms, including humans, and there is increasing evidence for intrinsic seasonal variation in immune systems. Changes are orchestrated through organisms' physiological clocks using cues such as day length. Ample research in diverse taxa has demonstrated multiple immune responses are modulated by photoperiod, but to date, there have been few experimental demonstrations that photoperiod cues alter susceptibility to infection. We investigated the interactions among photoperiod history, immunity and susceptibility in laboratory-bred three-spined stickleback (a long-day breeding fish) and its external, directly reproducing monogenean parasite *Gyrodactylus gasterostei*. We demonstrate that previous exposure to long-day photoperiods (PLD) increases susceptibility to infection relative to previous exposure to short days (PSD), and modifies the response to infection for the mucin gene *muc2* and Treg cytokine *foxp3a* in skin tissues in an intermediate 12 L : 12 D photoperiod experimental trial. Expression of skin *muc2* is reduced in PLD fish, and negatively associated with parasite abundance. We also observe inflammatory gene expression variation associated with natural inter-population variation in resistance, but find that photoperiod modulation of susceptibility is consistent across host populations. Thus, photoperiod modulation of the response to infection is important for host susceptibility, highlighting new mechanisms affecting seasonality of host–parasite interactions.

## 1. Introduction

Seasonality, whether hot–cold or wet–dry, is a prominent, ubiquitous and predictable environmental fluctuation necessitating substantial organismal plasticity. Seasonal change affects many aspects of an organism's environment, including host–parasite relationships and infection [1,2]. For example, relative resource limitation in winter restricts many physiological processes including immune responses; behavioural changes among hosts, such as aggregating while breeding, can affect transmission of parasites and pathogens among hosts [3]; and seasonal synchronization of development times [4] or increases in the abundance of a parasite or its vectors [5,6] modulates infection risk. Thus, seasonal variation can affect host immune responses, host–parasite associations and parasite virulence/abundance.

The timing of physiology to match seasons maximizes fitness, for example, by synchronizing reproduction and metabolism [7] with resource availability. The winter immunocompetence hypothesis (reviewed in [1]) suggests that immune defences are bolstered in winter due to increased pathogen risk and reduced

**Table 1.** Source populations for laboratory-bred fish. Parasite information sampled May 2013, see Magalhaes et al. [29].

| loch | location | G. arcuatus burden[a] | G. arcuatus prevalence | immune phenotype |
|------|----------|----------------------|------------------------|------------------|
| Chadha Ruaidh (SUS) | 57°36″ N; 7°12″ W | 0.0 ± 0.00 | 0.00 | susceptible (naive) |
| nan Strùban (RES) | 57°34″ N; 7°21″ W | 2.9 ± 1.08 | 0.69 | resistant |

[a]Mean ± s.e. for $n = 32$ (SUS) and 35 (RES) fish.

conflict for resources with somatic or reproductive processes within individuals [8]. However, seasonal immune variation is evident in non-seasonal breeders [9] and can be better explained by environment than reproductive condition when decoupled [10]. Research over the last 20 years has also demonstrated that it is both trait- and species-specific (reviewed in [6]). Seasonal immune modulation has been documented across diverse taxa, such as innate and inflammatory gene expression in humans [11] and three-spined stickleback (Gasterosteus aculeatus) [12], across innate, cell-mediated and humoural components in red-eared slider [13], species-specific innate immune traits of corals [14], and seasonal oscillations of within-population resistance and immune-associated SNPs in Drosophila [15].

As the most consistent signal of time of year, photoperiod is a common cue that allows organisms to maintain annual rhythm and align physiology with seasonality. Specifically, increases or decreases in day length relative to photoperiod history allow season to be determined year-round. Synchronizing physiology through photoperiod has been documented in humans and most vertebrates [16], and is achieved through a highly conserved hormonal system [17–19]. Much of the research on photoperiod modulation of seasonality has focused on mammals, which depend on incremental increases or decreases in melatonin secretion [20]. Melatonin, however, may be less significant for other seasonal physiological changes, such as reproduction, in non-mammals [6], but probably plays important roles in immune-mediation in fish [21]. Photoperiod and temperature are both principal seasonal cues in fish, the latter having a strong role in fish and other poikilotherms, and both are implicated in modulating seasonal phenotypes including reproduction, behaviour and immunity [19]. The effect of photoperiod on immunity has been extensively documented in Siberian hamsters (summarized in table S1 of [6]), but is also observed across diverse taxa including birds [22], fish [21,23] and plants [24].

Research focused on variable host responses to non-living pathogen mimetics, such as bacterial lipopolysaccharides [9,22,25] (LPS) and polyinosinic–polycytidylic acid (poly I : C), or proxies for resistance such as bacterial killing assays, have been integral in understanding mechanistic changes in host immunity. These, however, do not reveal whether these host modifications translate to changes in susceptibility under a full infection with a living parasite. Although work has suggested differing lethality of bacterial infections in short- and long-day hamsters [26], it is unclear how this may affect host–parasite associations [27].

We sought to address this gap in our knowledge using the three-spined stickleback and its naturally occurring monogenean ectoparasite, Gyrodactlyus gasterostei. Gyrodactylids infect the skin, fins and gills of hosts, reproducing clonally on host external surfaces without more complicated transmission requirements. Clonality can lead to high-density infections that become pathogenic [28], and secondary bacterial infection

is common. Stickleback exhibit extensive intra-species variation in many traits including immune responses [29] and display seasonal immunomodulation [12,30]. The role of intra-species variation in resistance and seasonality is a further area of research lacking empirical study [27]. Stickleback in the wild are annual long-day breeders, reproducing in late-spring, and typically live for 1–2 years, although this is variable among populations [29,31] and is extended in captivity (A.D.C.M. 2016, personal observation). Monogeneans, including Gyrodactylus sp., also display seasonal infection dynamics and modulation of reproduction rate [32].

We used laboratory-bred fish from the Scottish island of North Uist that vary greatly in their G. arcuatus prevalence and abundance, both in the wild [33] and in artificial infections [34]. We housed fish in contrasting photoperiods to induce changes in fish physiology, including immune responses, before conducting a controlled, factorial infection experiment with live parasites to examine the interplay between day length, natural resistance and susceptibility to a naturally occurring parasite.

We expected short-day treatments to reduce susceptibility under winter immunocompetence but expected variation among different components of immunity. Given gyrodactylids are seasonal, we also expected a stronger effect of day length in hosts bred from our naturally resistant population.

## 2. Material and methods

A single experiment was undertaken under laboratory conditions to investigate the interactions between photoperiod, immune responses and parasite susceptibility. All work involving animals was approved by the University of Nottingham ethics committee, and performed under UK Home Office Licence (PPL-40/3486).

### (a) Fish culturing and photoperiod treatment

Gravid females and reproductive males were collected from two lochs with contrasting Gyrodactylus arcuatus prevalence and resistance, Chadha Ruaidh (susceptible—SUS) and nan Strùban (resistant—RES) on North Uist, Scotland (table 1) in May 2014. SUS fish develop large parasite burdens due to evolutionary naivety to gyrodactylids, whereas the prevalence of G. arcuatus is relatively high in RES (table 1). Progeny were produced by in vitro crossing within each population following De Roij et al. [34], raised from eggs under laboratory conditions, and housed in family groups of equal density for 21 months, under six-month photoperiod regimes of 16 L : 8 D summers (February–July) and 8 L : 16 D winters (August–January) to simulate natural conditions (figure 1). At the beginning of month 22 (March 2016), 40 fish from each population were randomly assigned to short-day (maintained at 8 L : 16 D) or long-day photoperiod rooms (transitioned to 16 L : 8 D). This number was selected to maximize sample size within aquarium housing constraints. Each room was air-conditioned at 14°C, although temperatures fluctuated (16 L, mean max = 16.1°C, mean min = 13.8°C; 8 L, mean max = 15.4°C, mean min = 13.4°C).

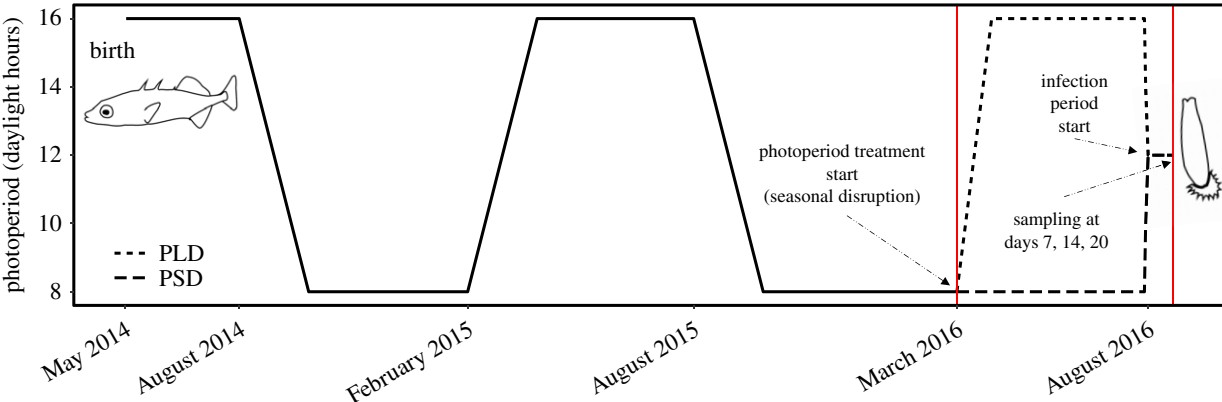

**Figure 1.** Experimental timeline tracking photoperiod changes through the lives of laboratory-reared F1s. Solid, vertical red lines denote the experimental photoperiod and infection period. (Online version in colour.)

Day length in the 16 L photoperiod room was steadily increased at 2 h per week from 8 to 16 h to more closely mimic natural transitions. Fish were housed individually in 10 l compartments of 20 l tanks for a total of 153 days.

Prior to infection, photoperiod in both treatments was changed at a rate of 30 min per day until an intermediate photoperiod (12 L : 12 D) was reached. This common intermediate photoperiod was necessary to ensure any photoperiod effects occurred through the host rather than the parasite, as seasonal diapause occurs in some parasites [4] and photoperiod moderates reproduction in other monogeneans [35]. In terms of time of year, this is analogous to a comparison between late winter (prior short days, 'PSD') and late summer (prior long days, 'PLD'). During the infection treatment, fish were housed for a further 20 days in a single 12 L room in individual 3 l tanks, with a 25% water change done every 3 days.

## (b) *Gyrodactylus* infections and sampling

A random 20 PSD and 15 PLD fish were infected from each photoperiod treatment, as nine PLD fish died during photoperiod treatment (electronic supplementary material, table S1). Elevated mortality of fish under 16 L : 8 D is normal in our experience of stickleback husbandry (A. Lowe and A.D.C.M. 2016, unpublished data). Fish were infected with *G. gasterostei*, collected from local wild stickleback. Our source populations are infected with *G. arcuatus*, thus the use of *G. gasterostei* removed possible effects of host local adaptation or host–parasite coevolution. *Gyrodactylus arcuatus* resistance is transferrable to *G. gasterostei* infections [34], and guppy hosts also lack local adaptation to gyrodactylids [36], consistent with transferrable host resistance.

Hosts were anaesthetized (280 mg l$^{-1}$ dose of MS222) and infected in a random order. To infect, the caudal fin was placed in a Petri dish under low-powered microscope in close contact with collected gyrodactylids until three individuals attached. Uninfected fish were similarly anaesthetized and handled to simulate infection protocol. While anaesthetized, we recorded the weight and standard length of all fish. Two days post-infection (dpi), fish that had stochastically lost their infections ($n = 3$) were re-infected, as clearing so soon is unlikely. At 7 and 20 dpi, all fish were anaesthetized, weighed and measured to track condition. Infected fish also had their parasite burdens observed and recorded blindly with no knowledge of treatment or source.

At 20 dpi, all fish were euthanized (400 mg l$^{-1}$ overdose of MS222) and had final measurements and parasite burdens recorded. The left operculum (skin) and spleen (immunologically relevant tissues [37]) were removed immediately and placed into RNAlater (Life Technologies). Tissue samples were stored at 4°C for 24 h and then −20°C prior to RNA extraction. We recorded sex, and weighed gonads, liver and adipose tissues. Subtracting these from total weight yielded somatic weight, a more stable assessment of weight. For proxies of condition, we took residual weight (against length), and adiposomatic index (ASI) and hepatosomatic index (HSI) as the ratio of tissue weight to somatic weight. These are common proxies of fish condition related to energy reserves. For reproductive condition of females, we calculated gonadosomatic index (GSI) and assessed males qualitatively based on testes and kidneys: 1 (small testes and kidney); 2 (enlarged, melanized testes, small kidney); 3 (enlarged, melanized testes and enlarged kidneys), following Robertson *et al.* [38]. Melanized testes are obvious during the breeding season, and fully mature males have enlarged kidneys due to production of the nest-building protein spiggin.

## (c) RNA isolation and qPCR

All qPCR work was conducted in accordance with the MIQE guidelines [39]. RNA was extracted from whole spleens and opercula in a random order and reverse-transcribed to cDNA following standard protocols (see electronic supplementary material for full details). qPCR reactions were performed using PrecisionFAST low ROX mastermix with SYBR green (Primerdesign) following standard protocols. Eight immune genes were selected to characterize different arms of the immune response and were identified based on previous studies in other fish and known roles of orthologous genes (electronic supplementary material, table S2; full details of gene choice in electronic supplementary material, methods), along with two housekeeping genes (*b2 m* and *rpl13a*) selected for normalization. cDNA from spleens and opercula of 69 individuals were amplified and relative expression values were calculated as per the ΔΔCq method [40] and adjusted for the amplification efficiencies of each primer pair. Expression values were standardized against the tissue-specific geometric mean Cq of two reference genes.

## 3. Results

## (a) Prior short days fish delayed reproductive maturation and were in better condition

Full GLM results and final models for all analyses are summarised in electronic supplementary material, table S3. Residual weight was modelled across the infection period as a function of days post-infection (0, 7, 14, 20), photoperiod treatment and infection treatment. Although residual weight deteriorated slightly in all fish during the first week at 12 l, it did not vary significantly over the course of the infection period (LMM, $F_{1,214} = 1.12$, $p = 0.290$; electronic supplementary material, figure S1A). Photoperiod was the major source of condition variation,

**Table 2.** *Gyrodactylus gasterostei* infection dynamics for infected fish from each treatment group.

| population | N | treatment | day 7 burden[a] | day 7 cleared[b] (%) | day 14 burden[a] | day 14 cleared[b] (%) | day 20 burden[a] | day 20 cleared[b] (%) | max burden[a] |
|---|---|---|---|---|---|---|---|---|---|
| Chadha Ruaidh | 10 | PSD (8 L : 16 D) | 3.20 ± 0.71 | 20 | 4.40 ± 1.24 | 20 | 7.70 ± 2.31 | 20 | 8.50 ± 2.26 |
| (SUS) | 8 | PLD (16 L : 8 D) | 4.00 ± 1.04 | 0 | 14.88 ± 3.82 | 0 | 41.50 ± 10.95 | 0 | 41.50 ± 10.95 |
| nan Strùban | 10 | PSD (8 L : 16 D) | 0.90 ± 0.38 | 40 | 1.40 ± 0.75 | 60 | 3.60 ± 2.85 | 60 | 4.80 ± 2.78 |
| (RES) | 7 | PLD (16 L : 8 D) | 4.29 ± 1.25 | 14.29 | 3.57 ± 1.09 | 14.29 | 5.57 ± 1.38 | 14.29 | 7.14 ± 1.08 |

[a]Values show mean ± s.e.
[b]Values represent the percentage of individuals with 0 parasites.

as PLD fish (mean residual weight = −0.051 ± 0.007) weighed relatively less than PSD fish (mean residual weight = 0.034 ± 0.010) across the infection period (LMM, $F_{1,69}$ = 13.32, $p <$ 0.001). Interestingly, infection by *G. gasterostei* had no significant effect on residual weight (LMM, $F_{1,69}$ = 3.05, $p$ = 0.088).

The first principal component of HSI and ASI (PC1 = 65.4% total variation, equal loadings) was used as a second measure of condition. As with residual weight, by this measure, PLD fish were in significantly poorer condition (PC1 mean = −0.237 ± 0.159) than PSD fish (PC1 mean = 0.189 ± 0.203), and condition was also worse in males (PC1 mean = −0.266 ± 0.240) than females (PC1 mean = 0.150 ± 0.159) (GLM, $F_{1,64}$ = 24.70, $p <$ 0.001; electronic supplementary material, figure S1B). PLD decrease in condition was worse in SUS fish (GLM, $F_{1,64}$ = 26.62, $p <$ 0.001; electronic supplementary material, figure S1B), largely because PSD-SUS fish (PC1 mean = 1.01 ± 0.229) were in the best condition (PLD-SUS PC1 mean = −0.231 ± 0.240, $p <$ 0.001, PSD-RES fish PC1 mean = −0.722 ± 0.191, $p <$ 0.001). Surprisingly, condition was greater in infected compared with uninfected SUS individuals (PC1 mean = 0.715 ± 0.292 versus 0.249 ± 0.252, $p$ = 0.008), but this was not true for RES fish (GLM, interaction between population and infection status, $F_{1,64}$ = 4.82, $p$ = 0.03).

PSD fish were in a more advanced reproductive condition at final sampling time. PSD females had significantly greater GSI (PSD-GSI mean = 75.3 ± 11.4; PLD-GSI mean = 35.9 ± 7.15), driven by differences within RES (PSD = 91.8 ± 16.8; PLD = 34.7 ± 14.3; $p <$ 0.001) (GLM, $F_{1,42}$ = 6.17, $p$ = 0.017; electronic supplementary material, figure S1C) and PSD males were more likely than chance to be reproductive ($\chi^2$ = 15.33, d.f. = 1, $p <$ 0.001; electronic supplementary material, figure S1D). However, during photoperiod treatments, PLD fish had been observed to be in reproductive condition (gravid females, red-throat males). Thus, PLD fish entered reproductive condition at the onset of photoperiod treatment, whereas PSD fish reproductive condition was delayed by extended short days (figure 1). We suggest then that, following photoperiod treatment and upon infection 12 L : 12 D, PLD fish reproductive condition declined and PSD fish entered reproductive condition due to the respective decrease/increase in photoperiod.

### (b) Prior long days treatment increases parasite susceptibility

Full details of infection dynamics can be found in table 2. Across the infection period, parasite abundances on PLD

fish were consistently higher than PSD fish (LMM, $F_{1,96}$ = 14.81, $p <$ 0.001; figure 2a). Further, parasite abundance increased at a greater rate in SUS fish compared with RES (LMM, $F_{1,96}$ = 9.86, $p$ = 0.002) and in males compared with females (LMM, $F_{1,96}$ = 4.14, $p$ = 0.042; figure 2a).

After 20 dpi, PLD fish harboured significantly greater parasite burdens (mean worm count = 24.7 ± 7.44) than those from PSD photoperiods (mean = 5.95 ± 1.83) (GLM, $F_{1,32}$ = 6.71, $p$ = 0.014; figure 2b). Unsurprisingly, susceptible-bred SUS fish harboured significantly greater abundances of parasites at 20 dpi (mean = 22.7 ± 6.33) than RES fish (mean = 4.76 ± 1.74) (GLM, $F_{1,32}$ = 8.57, $p$ = 0.006; figure 2b). Generally, the effect of photoperiod appears exaggerated in susceptible population fish; however, this interaction was not significant at the 5% level. Removal of fish that had been infected but cleared their infection did reveal a significant interaction between population and photoperiod (GLM, $F_{1,22}$ = 5.99, $p$ = 0.023), with clear differences between PLD-SUS (mean = 41.5 ± 11.00) and PSD-SUS (mean = 10.3 ± 2.71, $p <$ 0.001). By contrast, RES fish showed little difference in parasite abundance between PLD (mean = 6.5 ± 1.20) and PSD (mean = 8.4 ± 5.27, $p$ = 0.893) groups. These results suggest that PLD photoperiod treatments increased susceptibility in all fish; however in RES fish, the effect is driven by PSD fish clearing their infections rather than having lower abundances.

The likelihood of clearing infection over the course of the 20 days was also greater in PSD fish, although this result was not significant at the 5% threshold (GLM, $F_{1,33}$ = 3.90, $p$ = 0.057; figure 2c). Of the 35 fish infected, nine cleared the infection, eight of which had been housed under PSD conditions.

### (c) Immune gene expression is modified by photoperiod, population and infection

Expression of immune genes varied significantly by gene, tissue type and photoperiod (full summary in electronic supplementary material, table S4). Operculum *muc2* expression was particularly variable between photoperiod treatments, being significantly reduced in PLD fish (log₂-transformed relative expression ratio mean [L2RR] = −0.370 ± 0.097) compared with PSD fish (L2RR = −0.299 ± 0.099). There were also additional interactions with other treatment factors. SUS fish displayed a steeper decline in *muc2* expression following PLD treatment (GLM, $F_{1,61}$ = 10.22, $p$ = 0.002; figure 3a), driven by elevated expression in PSD-SUS (L2RR = 1.63 ± 0.339) compared with PLD-SUS (L2RR = −0.839 ± 0.292,

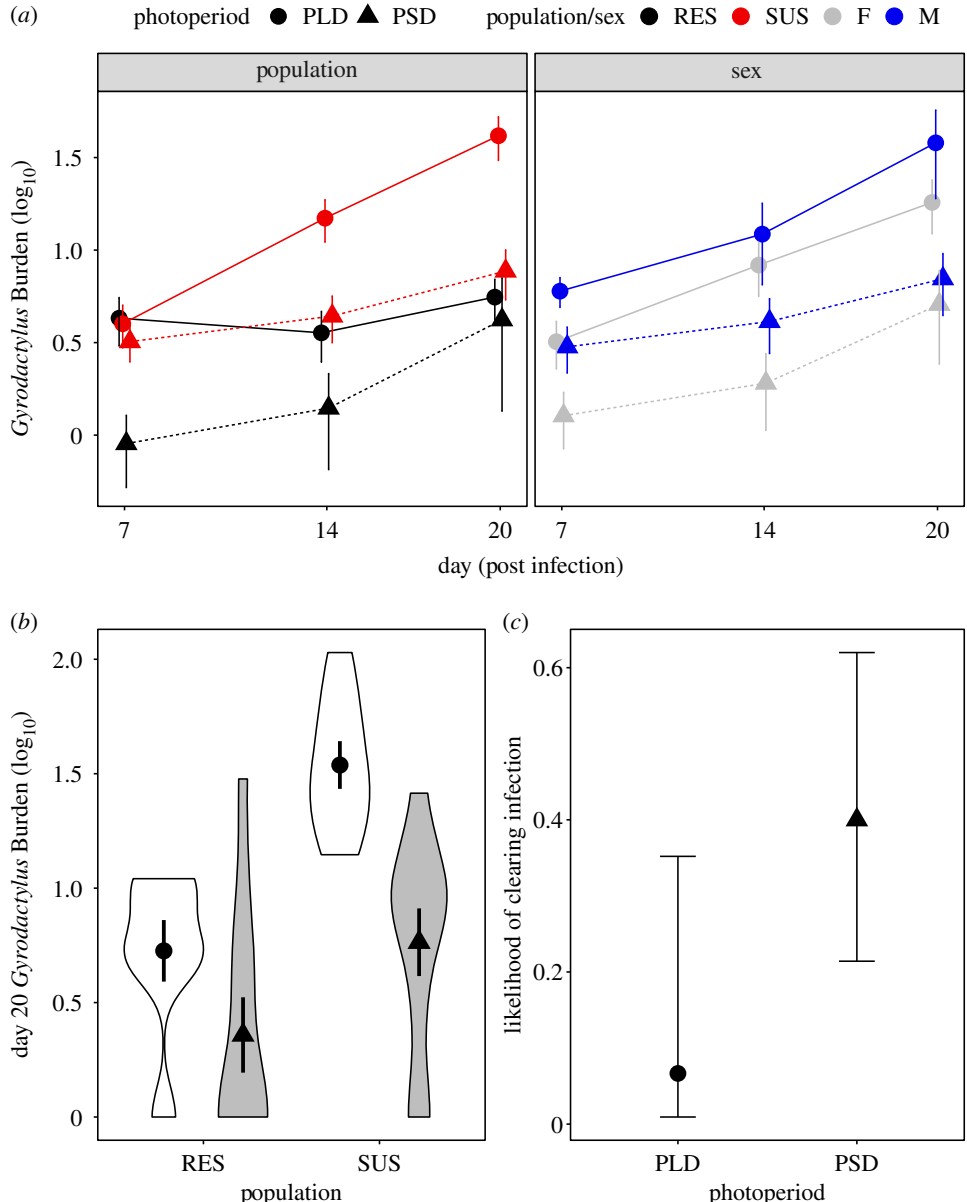

**Figure 2.** Increased parasite susceptibility under PLD treatments. (*a*) Mean ± s.e. *Gyrodactylus* burdens tracked over the infection period, grouped by significant model factors: population, sex and photoperiod treatment. (*b*) Log$_{10}$-transformed day 20 parasite burdens (+1), with group means ± s.e. (*c*) Likelihood of clearing infection, shown as the probability of clearing infection by day 20 with binomial error distributions for PSD and PLD fish. (Online version in colour.)

$p < 0.001$) and PSD-RES (L2RR = $-0.259 \pm 0.213$, $p < 0.001$), although expression was also significantly reduced in PLD-RES (L2RR = $-0.774 \pm 0.213$, $p = 0.019$) compared with PSD-RES. Males too displayed greater variation in *muc2* expression between PSD (L2RR = $1.84 \pm 0.39$) and PLD (L2RR = $-1.10 \pm 0.179$, $p < 0.001$) than did females (PSD L2RR = $-0.065 \pm 0.212$, PLD L2RR = $-0.684 \pm 0.179$, $p = 0.014$) (GLM, $F_{1,61} = 12.34$, $p < 0.001$; figure 3*a*).

Interestingly, infection of PSD-treated fish resulted in an upregulation of *muc2* in skin tissues (PSD-uninfected L2RR = $0.323 \pm 0.298$, PSD-infected L2RR = $1.30 \pm 0.411$, $p = 0.025$), while the opposite was observed in PLD fish (PLD-uninfected L2RR = $-0.268 \pm 0.167$, PLD-infected L2RR = $-1.38 \pm 0.236$, $p = 0.025$) (GLM, $F_{1,61} = 17.90$, $p < 0.001$; figure 3*a*). This suggests that photoperiod history not only modulates baseline *muc2* expression in skin tissues, but also affects the way in which expression of this gene responds to infection. Expression of *muc2* was much lower in spleen (electronic supplementary material, figure S2) than in skin (electronic supplementary material, figure S3).

Expression of *foxp3a* also varied in response to infection and differed according to photoperiod treatment. Infected PSD fish upregulated expression of this Treg cytokine while infected PLD fish reduced expression in the spleen (GLM, $F_{1,65} = 5.81$, $p = 0.019$; figure 3*b*) and particularly in skin tissues (GLM, $F_{1,65} = 11.85$, $p = 0.001$; figure 3*c*). *Post hoc* analyses revealed that within these interactions, there were significant downregulations in skin tissues of PLD fish in response to infection (uninfected L2RR = $-0.01 \pm 0.09$, infected L2RR = $-0.754 \pm 0.090$, $p < 0.001$), such that PLD-infected fish had significantly lower *foxp3a* expression than PSD-infected fish (PSD-infected L2RR = $-0.222 \pm 0.114$, $p = 0.020$). Again, this suggests that photoperiod treatment modifies the response to infection. We also observed a weak, but significant, pattern in skin tissues for *foxp3a* expression to be greater in PSD-SUS fish compared with PSD-RES fish, with the relationship inverted in PLD fish (GLM, $F_{1,63} = 4.18$, $p = 0.045$; figure 3*c*). These groups, however, did not differ significantly in *post hoc* analyses.

Photoperiod also affected the expression of Th1-associated genes in spleen tissues (*stat4* loading = 0.22; *tbet* loading = 0.98;

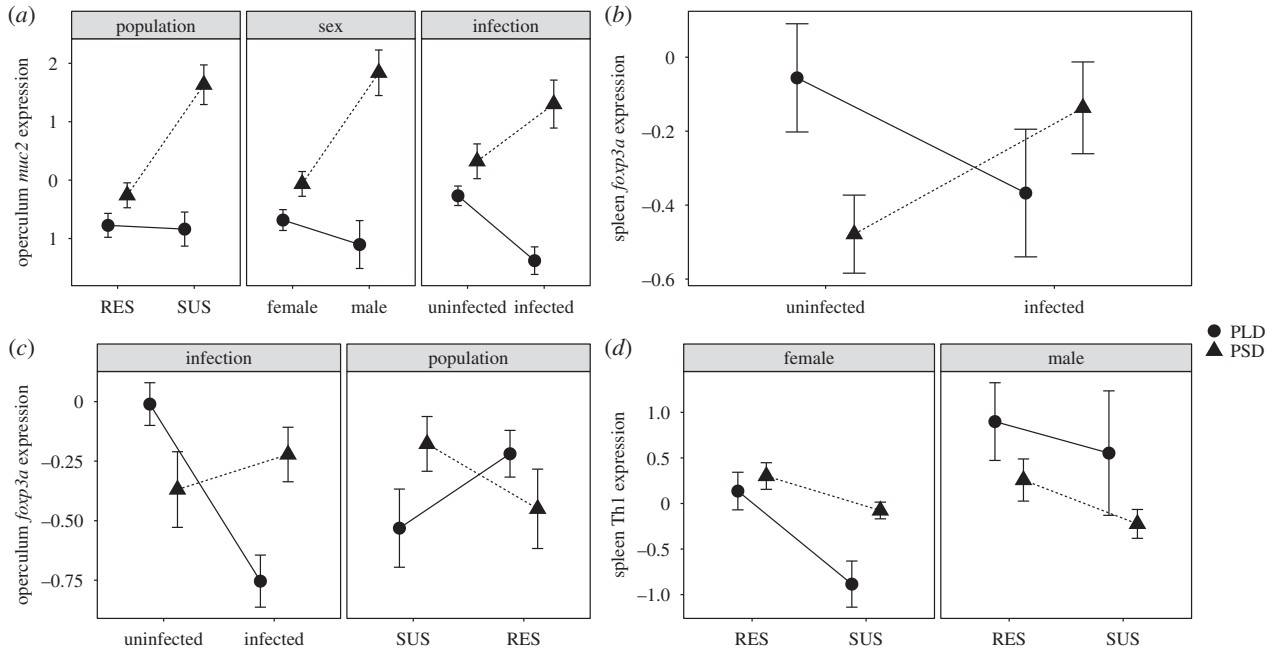

**Figure 3.** Photoperiod treatment affects expression of several genes in the spleen and opercula. Points show group means ± s.e. for log₂-relative expression ratios or PC1 of log₂-relative expression ratios. Genes with expression modified by photoperiod were *muc2* in the opercula (*a*), *foxp3a* in the spleen (*b*) and opercula (*c*), and a combined PC1 for Th1 genes *tbet* and *stat4* (*d*).

PC1 variation = 82.8%). Expression modulation of Th1-associated genes (predominantly *tbet*) was sex-dependent (figure 3*d*), and showed that PLD males upregulated expression while PLD females downregulated (GLM, $F_{1,65} = 4.31$, $p = 0.042$). This variation was significant for females (PSD PC1 mean = 0.390 ± 0.168, PLD PC1 mean = 0.561 ± 0.267, $p = 0.022$) but not for males (PSD PC1 mean = −0.001 ± 0.198, PLD PC1 mean = 0.437 ± 0.578, $p = 0.087$). Population-level differences in Th1-associated expression were also apparent in both spleen (GLM, $F_{1,65} = 8.31$, $p = 0.005$) and skin tissues (*stat4* loading = 0.58; *tbet* loading = 0.82; PC1 variation = 67.9%) (GLM, $F_{1,67} = 5.62$, $p = 0.021$). RES fish had significantly greater expression of Th1-associated genes in spleen tissues (PC1 mean = 0.395 ± 0.184), and lower expression in skin tissues (PC1 mean = −0.334 ± 0.215) than those bred from SUS (spleen PC1 mean = −0.362 ± 0.190; skin PC1 mean = 0.306 ± 0.167) (figure 3*d*).

Given the effects of photoperiod on condition and reproductive state, we remodelled immune variables that varied with photoperiod with the additional explanatory factors of GSI and condition PC1 scores. This allowed us to examine whether these variables better explain immune gene expression than our original discrete treatment factors, testing whether the former are part of the causal pathway. With these variables included, we retained our previous best-fitting models for spleen Th1 expression and spleen/operculum *foxp3a* expression. This suggests that photoperiod treatment affects gene expression directly through a mechanism that does not result from modification of reproductive and general condition. For operculum *muc2* expression, the interaction between photoperiod and population was replaced by an interaction between sex and condition PC1 (GLM, $F_{1,61} = 4.136$, $p = 0.046$). In this model, operculum *muc2* expression increased with increased condition, particularly in males. This suggests that operculum *muc2* expression is likely mechanistically linked to condition variation, although additional variation associated with photoperiod treatment (which was retained in models) is still important.

Th2-associated gene expression varied between source populations in the spleen (*stat6* loading = 0.51; *cmip* loading =

0.86; PC1 variation = 88.1%) (GLM, $F_{1,67} = 8.56$, $p = 0.005$) and skin (*stat6* loading = 0.61; *cmip* loading = 0.79; PC1 variation = 78.1%) (GLM, $F_{1,67} = 5.99$, $p = 0.017$) tissues. As observed for Th1-associated genes, Th2-associated expression was greater in RES fish for spleen tissues (PC1 mean = 0.457 ± 0.210) and lower in skin tissues (PC1 mean = −0.369 ± 0.204) compared with SUS (spleen PC1 mean = −0.419 ± 0.212; skin PC1 mean = 0.339 ± 0.204).

RES fish exhibited significantly increased splenic expression levels (L2RR = 0.705 ± 0.144; SUS L2RR = −0.431 ± 0.119) of the pro-inflammatory cytokine *tnfα* (GLM, $F_{1,66} = 45.35$, $p < 0.001$; electronic supplementary material, figure S2C), as did males (L2RR = 0.310 ± 0.212) compared against females (L2RR = −0.007 ± 0.131) (GLM, $F_{1,66} = 7.30$, $p = 0.009$). Conversely, in skin tissues, females (L2RR = −0.571 ± 0.082) exhibited greater *tnfα* expression than males (L2RR = −0.933 ± 0.092) (GLM, $F_{1,66} = 7.40$, $p = 0.008$; electronic supplementary material, figure S3D) due to reduced expression for males in skin tissue compared with the spleen. In addition, infected individuals (L2RR = −0.923 ± 0.091) had reduced *tnfα* expression in skin tissues compared with uninfected fish (L2RR = −0.510 ± 0.080) (GLM, $F_{1,66} = 10.91$, $p = 0.002$; electronic supplementary material, figure S3B). Similarly, infected individuals reduced expression of the Th17 transcription factor *rorc* in skin tissues (infected L2RR = −0.642 ± 0.135; uninfected L2RR = −0.073 ± 0.133) (GLM, $F_{1,67} = 8.98$, $p = 0.004$; electronic supplementary material, figure S3B). In spleen tissues, this gene was expressed more in males (L2RR = −1.46 ± 0.201) than females (L2RR = −2.06 ± 0.147) (GLM, $F_{1,66} = 6.14$, $p = 0.016$; electronic supplementary material, figure S2D).

## (d) Parasite burdens correlate with skin *muc2* expression

A correlation matrix including all immune variables and 20 dpi *G. gasterostei* burdens for infected individuals revealed collinearity between genes and final burdens (electronic supplementary material, figure S4A). The closest relationships

were generally tissue-specific, highlighting differential immune roles, and functional clustering of T-cell associated gene expression (Th1, Th2, *rorc*, *foxp3a*) for spleen tissues. *Gyrodactylus gasterostei* burdens were most strongly associated with decreased *muc2* ($R^2 = -0.41$) and *foxp3a* ($R^2 = -0.31$) expression in skin tissues and *tnfα* in spleen tissues ($R^2 = -0.24$). Modelling each immune variable by final parasite burdens demonstrated that fish with lower *muc2* expression in skin tissues harboured significantly greater infections (GLM, $F_{1,32} = 4.68$, $p = 0.038$; electronic supplementary material, figure S4B). Associations with all other immune variables were not significant (GLM, $0.001 \leq F_{1,32} \leq 2.84$, $0.102 \leq p \leq 0.977$).

## 4. Discussion

Here, we have experimentally confirmed a causal link between photoperiod and host susceptibility to a naturally occurring parasite in a controlled setting. Our results show that prior long-term exposure to long-day (PLD, 16 L : 8 D) photoperiods modified the expression of mucosal *muc2* in skin tissues and the Treg cytokine *foxp3a* in the spleen and skin tissues in response to infection with *G. gasterostei*. Prior short-day exposure (PSD, 8 L : 16 D) fish infected with *G. gasterostei* upregulated these genes and harboured lower peak infection burdens than PLD fish that downregulated expression. The manipulations conducted here thus demonstrate that seasonal physiology, moderated by photoperiod, modifies the host's response to infection and contributes to the observed variation in susceptibility between treatment groups. For the sake of discussion, we frame these results as PLD susceptibility because of accompanying deterioration in condition and status of stickleback as long-day breeders. However, this distinction is arbitrary, and results can also be considered as PSD-resistance or immunoenhancement.

Population-level differences in susceptibility persisted despite photoperiod treatment, along with differences between sexes. Modified reproductive timing is a probable cause of expression variation here, which seems likely given the controlling effect of changes in photoperiod [18], even across a single day [41]. Our PLD fish probably began sexual maturation shortly after photoperiod treatment began, while PSD fish had their maturation delayed until the infection stage of the experiment, which was housed under 12 l. This would also agree with the reproductive ecology of this species as long-day breeders, and the observed deterioration of condition in PLD fish and increased mortality following breeding in the wild [42]. Photoperiod-induced variation in susceptibility may, therefore, reflect direct interactions between day-length cues and the immune response or secondary interactions between photoperiod, condition and physiology, and susceptibility.

Seasonal patterns of disease are a common phenomenon across taxa [2,43,44], including stickleback [45], particularly around the breeding season [46]. Our results demonstrate that these patterns can occur through photoperiod-mediated annual rhythm in host susceptibility. This process is probably important alongside other explanations for seasonal parasitism such as seasonal changes in host behaviour [1,47], the local environment or climate [5]. PLD photoperiod treatment reduced baseline mRNA levels of our mucosal immunity marker, the Mucin-2 gene *muc2*, in skin tissues of all PLD fish and reversed the pattern of upregulation observed in PSD fish in response to infection. Mucin-2 is an important

constituent of mucosal secretions, and in mammals helps separate pathogens and commensals in epithelia [48]. Mucosal immunity has been shown to vary seasonally in other vertebrates [49] including fish [50,51]. Mucosal immunity, a cocktail of cellular innate, humoral innate and adaptive immune components, is particularly important in an aquatic environment [52], and for fish managing gyrodactylid infections [53]. For instance, *G. cichlidarum* aggregate away from tilapia host fins with high mucous cell density [54], and inhibition of mucous cell discharge increases salmon susceptibility to *G. derjavani* [55]. Upregulation of *muc2* in infected PSD fish is consistent with an immune response against gyrodactylid infection, but downregulation in infected PLD fish suggests that these individuals are unable to mount the necessary response.

Mechanistically, mucosal immunity may be modulated by general condition; starved blue catfish, for example, downregulate *mucin-5AC-like* and *mucin-2-like* in skin tissues [56]. In our results, general condition was a good predictor of skin *muc2* expression. Photoperiod treatment was also retained in these models however, explaining variation in expression unaccounted for by condition. Because modified mucosal immunity may feasibly be a consequence of reduced condition as much as an adaptive response to it, collinearity is a concern. Our study design lacks the statistical power to distinguish between such hypotheses, but increased sample sizes in future studies would permit the use of analyses such as structural equation modelling to disentangle collinearity.

With the exception of *muc2* expression in skin tissues, none of our genes exhibited consistent expression changes in comparisons of all PLD and all PSD fish. This finding agrees with the results of a recent study in sticklebacks that suggested photoperiod modulation in a laboratory environment is insufficient to yield expression changes observed in wild individuals [30]. We did, however, observe photoperiod-specific responses to infection for both *muc2* and *foxp3a*. The use of laboratory-reared individuals here compared with wild individuals brought into the laboratory is also expected to modify results, and our use of a novel experimental infection treatment clearly demonstrates the significant role of day-length cues in response to infection, and ultimately susceptibility.

Photoperiod is a significant cue for timing of breeding [20], the co-expression of immune responses with seasonal changes to sex hormones modulates immunity [57], and reproductive hormones induce sex-specific immune responses in humans [58]. PSD fish were entering reproductive condition, which likely increased blood plasma levels of the immunosuppressive [59] androgen 11-ketotestosterone and may explain observed declines in PSD male Th1 expression. Gonadectomy studies in Siberian hamsters have documented reduced strength of photoperiod effects with castration, but castration does not eliminate effects of day length [25]. Similarly, we found photoperiod strongly influenced reproductive physiology, but specific measures such as GSI were poor at explaining expression and infection variation.

Expression of splenic *tnfα* and Th1 and Th2-associated genes varied between populations, but, with the exception of splenic Th1 genes, were unaffected by photoperiod. Both *tnfα* and Th1 expression (particularly pro-inflammatory *tbet*) may explain varying natural resistance to gyrodactylids, as baseline expression was greater in our naturally resistant population. Expression of inflammatory genes *tnfα* and *il-β* is important in the initial stages of *G. derjavini* infection in rainbow trout

[60], and expression of these genes also increases with *G. arcuatus* burdens in wild stickleback from Scotland [61]. Importance in the early stages of infection may explain why we fail to observe associations between inflammatory expression and infection burdens when sampling after 20 days.

The biological mechanisms of internal time-keeping can dampen with age, at least for circadian rhythm [62], and our laboratory-bred fish were old (greater than 2 years) relative to wild lifespans (1–2 years), although captive stickleback can live up to 5 years. That we observe photoperiod effects despite this suggests the observed variation between treatments may be a conservative estimate.

The ability to assess the effect of photoperiod alongside intraspecific variation in susceptibility is a novel element of this study, allowing us to address open questions in the field [27]. We found no evidence to support the hypothesis that photoperiod-induced immune changes are adaptive responses to seasonal parasite prevalence as effects were apparent in both populations, despite susceptible fish being naive to gyrodactylids. However, natural parasite communities on North Uist are diverse and complex [33,63], thus seasonal host immunity may be adaptive for other parasites with greater fitness implications. Both *Schistocephalus solidus* and *Diplostomum gasterostei* are prevalent across North Uist, exhibit some seasonality [45] and incur fitness costs through increased mortality/predation risk [64,65] and reduced reproduction [66].

The decision to examine PLD and PSD treatments, as opposed to infected fish during long/short-day treatment, was necessary to ensure infection variation occurred through hosts and not interactions between photoperiod and gyrodactylids. This design, however, limits our ability to say with certainty whether susceptibility differences extend to hosts held in contrasting 16 L : 8 D and 8 L : 16 D photoperiods. In nature, that an individual's current physiological condition is a product of its photoperiod history is a cornerstone of circannual rhythm theory, so such a distinction is not strictly important for our conclusions here. However, future studies employing systems with a living parasite with a confirmed lack of photoperiod-virulence effects would be important additions to pin down mechanistic interactions between current photoperiod and susceptibility.

In conclusion, we have demonstrated that experimental modulation of photoperiod can increase susceptibility to natural parasites, due to modulating host physiology, condition and immune responses to infection. Seasonal host–parasite interactions are poorly understood [27]; however, here we provide evidence showing that while seasonal changes to hosts increase susceptibility, host seasonality can occur irrespective of seasonal parasitism. The neuroendocrine processes controlling photoperiod modulation are well conserved [18], and so these results have implications for understanding seasonal host–parasite interactions across taxa, including humans. Our results indicate that seasonal host immune responses may be an important consideration for understanding seasonal infection and disease epidemiology. Further, photoperiod is an important aspect of industries such as aquaculture, where these findings will be directly applicable in understanding parasitism of stocks.

**Ethics.** All work involving animals was approved by the University of Nottingham ethics committee, and performed under UK Home Office Licence (PPL-40/3486).

**Data accessibility.** Raw infection experiment results; measures of condition; log-transformed relative expression ratios; multivariate gene expression PCs and analysis R script are available from the Dryad Digital Repository: https://doi.org/10.5061/dryad.dz08kprv7 [67].

**Authors' contributions.** J.R.W. and A.D.C.M. planned the experiment. J.R.W. and M.A.M. planned and performed the artificial infections. J.R.W. performed molecular biology, statistics and writing. J.E.B. provided support and information regarding immunology. A.D.C.M. and J.E.B. contributed towards writing.

**Competing interests.** The authors declare no competing interests.

**Funding.** This work was supported by the Biotechnology and Biological Sciences Research Council (grant number BB/J014508/1). The BBSRC DTP studentship was completed by J.R.W.

**Acknowledgements.** The authors wish to acknowledge Abdul Rahman Singkam, Anne Lowe and Alan Crampton for their care and assistance in the aquarium. Francis Ebling provided insight and knowledge regarding the interactions between photoperiod and neurology. We also thank four anonymous reviewers for helpful comments.

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
