## [Reviewer comments · Proceedings of the Royal Society B: Biological Sciences]

Review History

RSPB-2019-2093.R0 (Original submission)

Review form: Reviewer 1

Recommendation

Major revision is needed (please make suggestions in comments)

Scientific importance: Is the manuscript an original and important contribution to its field?

Good

General interest: Is the paper of sufficient general interest?

Excellent

Quality of the paper: Is the overall quality of the paper suitable?

Good

Is the length of the paper justified?

Yes

Should the paper be seen by a specialist statistical reviewer?

No

Do you have any concerns about statistical analyses in this paper? If so, please specify them explicitly in your report.

No

It is a condition of publication that authors make their supporting data, code and materials available - either as supplementary material or hosted in an external repository. Please rate, if applicable, the supporting data on the following criteria.

Is it accessible?

Yes

Is it clear?

Yes

Is it adequate?

Yes

Do you have any ethical concerns with this paper?

No

Comments to the Author

The manuscript by Whiting et al. investigated the interactions among photoperiod, immunity and disease susceptibility in a fish species. Specifically, to examine the effects of photoperiod on disease, they lab-reared fish derived from wild populations of three-spined stickleback in either long summer-like photoperiod or short winter-like photoperiods. They then experimentally infected them with the parasite *Gyrodactylus gasterostei* and examined disease clearance, as well as gene expression of inflammatory genes in skin. They showed that long-day housed fish were less capable in clearing parasitic infections and had reduced expression of skin mucin gene (*muc2*) expression and Treg cytokine *fox3a* expression. While there were differences in inflammatory gene expression across populations of fish, photoperiod modulation of infection was consistent across host populations. The authors' suggest these findings demonstrate that photoperiodic modulation of immunity is important for seasonal changes in host susceptibility

Major Issues

One of the major issues, which the authors do admit and address, is whether changes in skin immunity are the cause or consequence of infection. It is quite possible, even likely, that inflammatory responses would be elevated in response to experimental infections. As I said, the author acknowledge this and indicate that their does not does not allow disentangling this relationship; that's said it does weaken my enthusiasm about the putative mechanisms underlying disease resistance in this study.

The design of this study has all individuals being transferred to LD 12:12 following photoperiod manipulation so that they are in a "common photoperiod." While seemingly reasonable on face value, prior photoperiodic conditions drive an animal's response to future day lengths such that a subsequent photoperiod could be perceived as long or short relative to the prior photoperiodic history. 12:12 is particularly troublesome in that it is considered an intermediate photoperiod for many vertebrates and interpreted as long vs short depending on individual. It would have probably been more logical to keep animals in their respective conditions; this would not allow separation of current vs. prior photoperiod, but would avoid the confound of prior photoperiodic history.

This last point is more philosophical than procedural. There is a robust literature in vertebrates, especially mammals and birds, looking at seasonal/photoperiodic changes in immune response,

and in many of these animal, they show short-day immunoenhancement relative to long days. One of the hypotheses proffered is the winter immunoenhancement hypothesis by which animal have evolved to enhance immunity in short days to buffer against winter stressors. In fact, some causal mechanisms, cortisol, melatonin, adipokines have been suggested. Given this, why did the author decide to couch their findings as long-day suppression rather than short-day enhancement. Admittedly, its arbitrary as to which photoperiod is “baseline” but it struck me as unusual give the previous literature in their field.

Minor Issues

Abstract and intro. While I agree that there are few to no studies looking at photoperiodic changes in susceptibility to infection, despite the ample literature with respect to immune measures, I think its work acknowledging some studies that utilize proxies of infection (e.g., LPC, PolyIC) which are bacterial or viral memetics, or ex vivo “infection resistance” in the form of ex vivo bacterial killing assays. While these studies do not test actual infection susceptibility with the organism, that have contributed to the seasonal disease literature. Related, I would not agree that seasonality of immune responses generally manifests through changes in innate immunity; there are abundant examples of changes in both humoral and cellular immunity as well.

Abstract, line 29. Change between to among since it’s a comparison of more than two items.

Given the design of the experiment and the use of F1 animal from susceptible and resistant populations, do all of the experimental animals start out parasite-free?

Why are animal re-infected after two days? In other words, if resistance to infection is one of the variables on interest, why is the lack of infection two days later not simply considered as part of this resistance? Is it owing to the time course of putative clearance?

I understand the desire to call the photoperiods summer photoperiod and winter photoperiod. But because photoperiod and not seasonality was tested her, I suggest using the more standard long-day and short-day as it can be confusing to some that actual seasonality was not assessed in this study.

The reproductive response were counterintuitive to me; is this species short-day breeders?

Review form: Reviewer 2

Recommendation

Major revision is needed (please make suggestions in comments)

Scientific importance: Is the manuscript an original and important contribution to its field?

Good

General interest: Is the paper of sufficient general interest?

Good

Quality of the paper: Is the overall quality of the paper suitable?

Good

Is the length of the paper justified?

Yes

Should the paper be seen by a specialist statistical reviewer?

No

Do you have any concerns about statistical analyses in this paper? If so, please specify them explicitly in your report.

No

It is a condition of publication that authors make their supporting data, code and materials available - either as supplementary material or hosted in an external repository. Please rate, if applicable, the supporting data on the following criteria.

Is it accessible?

No

Is it clear?

N/A

Is it adequate?

N/A

Do you have any ethical concerns with this paper?

No

Comments to the Author

In this manuscript, the authors demonstrated that changes in photoperiod affected parasitic infection dynamics, reproductive physiology, body condition, and immune gene expression in two populations of stickleback that varied in infection susceptibility. This manuscript highlights unique experimental data that will further our understanding of environmental and physiological regulation of immunity and parasite dynamics in a non-model system. Overall, I found the study compelling and applaud the authors' use of a naturally occurring host-parasite system.

General Comments:

1. Throughout the manuscript, I found the scope of the literature cited very narrow, primarily focusing on fish and humans. I would recommend citing more of the mammalian, bird, and even plant literature on seasonality in immune function and disease (e.g., Ashley et al. 2013 (JEB), Fonseca et al. 2014, PLOS ONE).
2. I would suggest restructuring the introduction to improve the logical flow of ideas. For example, the first paragraph jumps from talking about fur color to disease to immunity with minimal connections between these ideas. I suggest starting with 1) a broad discussion of seasonal variation in environmental variables (perhaps including seasonality/variation in disease risk and host susceptibility) and then discussing 2) how organisms frequently use photoperiod to time their physiological systems (e.g, reproduction, immunity) to best match the environment to maximize fitness. Finally, since this is an experimental study, I'd recommend including some brief hypotheses and predictions for your study based on the published literature.
3. I'd recommend incorporating brief life history/ecological information about the stickleback to aid data interpretation. For example, when do wild stickleback normally reproduce? Since your data found that winter sticklebacks are more likely to be sexually mature, this seems like something relevant to discuss if this is an abnormal physiological result. In addition to providing life history on the stickleback, it would be useful to know a bit more about the parasite itself and its pathogenicity to evaluate the potential selective pressures on the host.
4. For the figure captions, I'd recommend making them a bit more descriptive e.g., "experimental outcomes" is vague.
5. Given the multitude of immune genes and tissues that were sampled, I'd suggest adding a small table that summarizes the "effect" columns in Tables S3 and S4 to make it easier for the reader to keep track of the major results.

Specific Comments:

Line 51: "Synchronising temporal variation through photoperiod", do you mean synchronizing physiology?

Lines 54-55: Since you don't measure or discuss thyroid hormone in the rest of the manuscript, I was distracted by this specific explanation of the mechanism involved.

Line 58: Please provide a citation for this sentence.

Line 60: In Line 58, it is mentioned that most of immune seasonality is manifested through changes in innate immunity, but this example in stickleback (adaptive T-cell response), is not within innate immunity. Perhaps rephrase first sentence to reflect this?

Line 68: "known from controlled studies" – can you provide which species these were done in?

Lines 154-156: Please provide a sample size for the number of fish that had to be re-infected because previous infection didn't take (particularly since this seemed to require additional handling/manipulation).

Lines 173-175: Please provide a citation that justifies using the measure of reproductive condition (REPRO) in males. Also, how accurate is this measure in determining actual reproductive condition in males?

Line 192: I found the term "post-reproductive" confusing.

Line 324: Since you're now discussing potential effects of condition and reproductive state, perhaps a new subheading here?

Line 386: If photoperiod drives sexual maturation in these fish, then why are the winter fish more sexually mature than the summer fish? I am also unclear about the mechanism explaining why summer fish are in worse condition than winter fish. Are there other infections/physiological changes that take place in the summer that could explain this decline in condition? Is this an artifact of captivity?

Lines 416-425: I acknowledge you can't determine the exact relationship between muc2 expression and condition given your data, but could you speculate given the literature about why these mucin genes decrease in fish in lower condition? Is this part of the stress response in these fish?

Line 439: Replace "gender-specific" with "sex-specific"

Lines 473-476: Can you specify some of these other commonly occurring seasonal parasites/diseases? Do any of these affect the fish more severely than *G. gasterostei*?

Decision letter (RSPB-2019-2093.R0)

24-Oct-2019

Dear Dr Whiting:

I am writing to inform you that your manuscript RSPB-2019-2093 entitled "Summertime blues: Long day length alters immune responses and increases susceptibility to parasitic infection in stickleback" has, in its current form, been rejected for publication in Proceedings B.

This action has been taken on the advice of referees, who have recommended that substantial revisions are necessary. With this in mind we would be happy to consider a resubmission, provided the comments of the referees are fully addressed. However please note that this is not a provisional acceptance.

Sincerely,
 Professor Hans Heesterbeek
 mailto: proceedingsb@royalsociety.org

Associate Editor

Board Member: 1

Comments to Author:

This manuscript has been evaluated by two expert reviewers, who both agree that the paper presents interesting results. However, both reviewers also point to number of issues that need to be addressed. In particular, both reviewers remark on the currently narrow presentation of the background literature and prevailing hypotheses on seasonality/photoperiod and immune function in vertebrates. In addition, R2 suggests that life history and ecological information about the study species would facilitate data interpretation, while R1 identifies potential flaws in the study design that need further clarification and/or justification. Both reviews also make a number of other suggestions that should be fully considered.

Reviewer(s)' Comments to Author:

Referee: 1

Comments to the Author(s)

The manuscript by Whiting et al. investigated the interactions among photoperiod, immunity and disease susceptibility in a fish species. Specifically, to examine the effects of photoperiod on disease, they lab-reared fish derived from wild populations of three-spined stickleback in either long summer-like photoperiod or short winter-like photoperiods. They then experimentally infected them with the parasite *Gyrodactylus gasterostei* and examined disease clearance, as well as gene expression of inflammatory genes in skin. They showed that long-day housed fish were less capable in clearing parasitic infections and had reduced expression of skin mucin gene (*muc2*) expression and Treg cytokine *fox3a* expression. While there were differences in inflammatory gene expression across populations of fish, photoperiod modulation of infection was consistent across host populations. The authors' suggest these findings demonstrate that photoperiodic modulation of immunity is important for seasonal changes in host susceptibility

Major Issues

One of the major issues, which the authors do admit and address, is whether changes in skin immunity are the cause or consequence of infection. It is quite possible, even likely, that inflammatory responses would be elevated in response to experimental infections. As I said, the author acknowledge this and indicate that their does not does not allow disentangling this

relationship; that's said it does weaken my enthusiasm about the putative mechanisms underlying disease resistance in this study.

The design of this study has all individuals being transferred to LD 12:12 following photoperiod manipulation so that they are in a "common photoperiod." While seemingly reasonable on face value, prior photoperiodic conditions drive an animal's response to future day lengths such that a subsequent photoperiod could be perceived as long or short relative to the prior photoperiodic history. 12:12 is particularly troublesome in that it is considered an intermediate photoperiod for many vertebrates and interpreted as long vs short depending on individual. It would have probably been more logical to keep animals in their respective conditions; this would not allow separation of current vs. prior photoperiod, but would avoid the confound of prior photoperiodic history.

This last point is more philosophical than procedural. There is a robust literature in vertebrates, especially mammals and birds, looking at seasonal/photoperiodic changes in immune response, and in many of these animal, they show short-day immunoenhancement relative to long days. One of the hypotheses proffered is the winter immunoenhancement hypothesis by which animal have evolved to enhance immunity in short days to buffer against winter stressors. In fact, some causal mechanisms, cortisol, melatonin, adipokines have been suggested. Given this, why did the author decide to couch their findings as long-day suppression rather than short-day enhancement. Admittedly, its arbitrary as to which photoperiod is "baseline" but it struck me as unusual give the previous literature in their field.

Minor Issues

Abstract and intro. While I agree that there are few to no studies looking at photoperiodic changes in susceptibility to infection, despite the ample literature with respect to immune measures, I think its work acknowledging some studies that utilize proxies of infection (e.g., LPC, PolyIC) which are bacterial or viral memetics, or ex vivo "infection resistance" in the form of ex vivo bacterial killing assays. While these studies do not test actual infection susceptibility with the organism, that have contributed to the seasonal disease literature. Related, I would not agree that seasonality of immune responses generally manifests through changes in innate immunity; there are abundant examples of changes in both humoral and cellular immunity as well.

Abstract, line 29. Change between to among since it's a comparison of more than two items.

Given the design of the experiment and the use of F1 animal from susceptible and resistant populations, do all of the experimental animals start out parasite-free?

Why are animal re-infected after two days? In other words, if resistance to infection is one of the variables on interest, why is the lack of infection two days later not simply considered as part of this resistance? Is it owing to the time course of putative clearance?

I understand the desire to call the photoperiods summer photoperiod and winter photoperiod. But because photoperiod and not seasonality was tested her, I suggest using the more standard long-day and short-day as it can be confusing to some that actual seasonality was not assessed in this study.

The reproductive response were counterintuitive to me; is this species short-day breeders?

Referee: 2

Comments to the Author(s)

In this manuscript, the authors demonstrated that changes in photoperiod affected parasitic infection dynamics, reproductive physiology, body condition, and immune gene expression in

two populations of stickleback that varied in infection susceptibility. This manuscript highlights unique experimental data that will further our understanding of environmental and physiological regulation of immunity and parasite dynamics in a non-model system. Overall, I found the study compelling and applaud the authors' use of a naturally occurring host-parasite system.

General Comments:

1. Throughout the manuscript, I found the scope of the literature cited very narrow, primarily focusing on fish and humans. I would recommend citing more of the mammalian, bird, and even plant literature on seasonality in immune function and disease (e.g., Ashley et al. 2013 (JEB), Fonseca et al. 2014, PLOS ONE).
2. I would suggest restructuring the introduction to improve the logical flow of ideas. For example, the first paragraph jumps from talking about fur color to disease to immunity with minimal connections between these ideas. I suggest starting with 1) a broad discussion of seasonal variation in environmental variables (perhaps including seasonality/variation in disease risk and host susceptibility) and then discussing 2) how organisms frequently use photoperiod to time their physiological systems (e.g, reproduction, immunity) to best match the environment to maximize fitness. Finally, since this is an experimental study, I'd recommend including some brief hypotheses and predictions for your study based on the published literature.
3. I'd recommend incorporating brief life history/ecological information about the stickleback to aid data interpretation. For example, when do wild stickleback normally reproduce? Since your data found that winter sticklebacks are more likely to be sexually mature, this seems like something relevant to discuss if this is an abnormal physiological result. In addition to providing life history on the stickleback, it would be useful to know a bit more about the parasite itself and its pathogenicity to evaluate the potential selective pressures on the host.
4. For the figure captions, I'd recommend making them a bit more descriptive e.g., "experimental outcomes" is vague.
5. Given the multitude of immune genes and tissues that were sampled, I'd suggest adding a small table that summarizes the "effect" columns in Tables S3 and S4 to make it easier for the reader to keep track of the major results.

Specific Comments:

Line 51: "Synchronising temporal variation through photoperiod", do you mean synchronizing physiology?

Lines 54-55: Since you don't measure or discuss thyroid hormone in the rest of the manuscript, I was distracted by this specific explanation of the mechanism involved.

Line 58: Please provide a citation for this sentence.

Line 60: In Line 58, it is mentioned that most of immune seasonality is manifested through changes in innate immunity, but this example in stickleback (adaptive T-cell response), is not within innate immunity. Perhaps rephrase first sentence to reflect this?

Line 68: "known from controlled studies" – can you provide which species these were done in?

Lines 154-156: Please provide a sample size for the number of fish that had to be re-infected because previous infection didn't take (particularly since this seemed to require additional handling/manipulation).

Lines 173-175: Please provide a citation that justifies using the measure of reproductive condition (REPRO) in males. Also, how accurate is this measure in determining actual reproductive condition in males?

Line 192: I found the term "post-reproductive" confusing.

Line 324: Since you're now discussing potential effects of condition and reproductive state, perhaps a new subheading here?

Line 386: If photoperiod drives sexual maturation in these fish, then why are the winter fish more sexually mature than the summer fish? I am also unclear about the mechanism explaining why summer fish are in worse condition than winter fish. Are there other infections/physiological changes that take place in the summer that could explain this decline in condition? Is this an artifact of captivity?

Lines 416-425: I acknowledge you can't determine the exact relationship between muc2 expression and condition given your data, but could you speculate given the literature about why

these mucin genes decrease in fish in lower condition? Is this part of the stress response in these fish?

Line 439: Replace "gender-specific" with "sex-specific"

Lines 473-476: Can you specify some of these other commonly occurring seasonal parasites/diseases? Do any of these affect the fish more severely than *G. gasterostei*?

Author's Response to Decision Letter for (RSPB-2019-2093.R0)

See Appendix A.

RSPB-2020-0288.R0

Review form: Reviewer 3

Recommendation

Reject – article is scientifically unsound

Scientific importance: Is the manuscript an original and important contribution to its field?

Good

General interest: Is the paper of sufficient general interest?

Good

Quality of the paper: Is the overall quality of the paper suitable?

Marginal

Is the length of the paper justified?

Yes

Should the paper be seen by a specialist statistical reviewer?

No

Do you have any concerns about statistical analyses in this paper? If so, please specify them explicitly in your report.

No

It is a condition of publication that authors make their supporting data, code and materials available - either as supplementary material or hosted in an external repository. Please rate, if applicable, the supporting data on the following criteria.

Is it accessible?

No

Is it clear?

N/A

Is it adequate?

N/A

Do you have any ethical concerns with this paper?

No

Comments to the Author

This study examines the effect of photoperiod upon immunity and susceptibility to parasitism in three-spined stickleback. Many studies have reported upon seasonal changes in immune function, but very few have employed the use of real parasites and their effect upon host susceptibility. Overall, this study is generally well-presented, although I have some concerns regarding experimental design and presentation of methods that should be addressed.

Major points:

1.) Experimental design- In general, the methods are quite confusing. Can they be simplified? Or at least can an experimental timeline be produced? If I understand the set-up of the experiment correctly, there were two photoperiod treatments, and the LD group were exposed to increasing photoperiods (2 h of light per week) up to 16L:8D, and the 8L:16D groups remained the same. Then, the authors shift photoperiods so that both groups are at 12L:12D, so that groups can be infected by monogeneans experiencing the same photoperiod. The problem with this approach is that SD fish will be photostimulated on 12L:12D and are thus no longer in "winter" condition, and thus are no longer 8L. The LD may still be in reproductive condition, but will be progressing to a non-breeding state because of decreasing photoperiod. Therefore, I argue that this approach confounds assignment of fish to LD or SD. After 20 days of LD exposure (in SD fish), the HPG axis will be fully stimulated. This is corroborated by the SD fish being in more advanced reproductive condition than LD fish in the results. Instead the authors should have probably collected monogeneans from fish that were exposed to 12L:12D, and then transferred to the 8L and 16L groups. Given the present experimental design, what the authors are really testing is whether photoperiod history has an effect upon host susceptibility. I suppose the paper could be couched in those terms...

2.) Methods- In general, the paper should be re-written so that the methods are more clear. As stated above, an experimental timeline here would be helpful

Minor comments:

Abstract-You never state in the abstract if stickleback are short- or long-day breeders.

Line 73. Delete "appears"

Line 136. Please provide evidence that rapid photoperiod shifts are "stressful" in these fish. Previous studies have rapidly shifted photoperiod, and there seem to be no stress-induced effects.

Line 250-252- And herein lies the confound in this study. How can you term fish SD, when they are in breeding condition?

Line 286-287. Why mention this result if it does not meet the 5% threshold then?

Review form: Reviewer 4 (Mary L. Westwood)**Recommendation**

Major revision is needed (please make suggestions in comments)

Scientific importance: Is the manuscript an original and important contribution to its field?
Good

General interest: Is the paper of sufficient general interest?
Excellent

Quality of the paper: Is the overall quality of the paper suitable?
Good

Is the length of the paper justified?
Yes

Should the paper be seen by a specialist statistical reviewer?
No

Do you have any concerns about statistical analyses in this paper? If so, please specify them explicitly in your report.
No

It is a condition of publication that authors make their supporting data, code and materials available - either as supplementary material or hosted in an external repository. Please rate, if applicable, the supporting data on the following criteria.

Is it accessible?
Yes

Is it clear?
Yes

Is it adequate?
Yes

Do you have any ethical concerns with this paper?
No

Comments to the Author

The manuscript by Whiting et al. uses stickleback from susceptible and naïve populations to investigate how long and short-day photoperiods affect parasitic infection dynamics. The authors should be commended for the use of a non-model organism and a naturally occurring parasite. There are some areas in which the author still needs to improve, especially chronobiology wording and its use in presenting results. However, I think these changes should be easy to fix and implement, and don't detract from the novelty of the results. Overall, the work was well executed and presents a valuable contribution to the understanding of the effect of photoperiod on parasitic infection dynamics.

Major comments:

1. Chronobiology terminology is often difficult to accurately use, and I worry some of the usage here may be misleading. For example, the paper argues for differences between LD and SD fish, which is driven by photoperiod manipulations. However, it would be more accurate to claim that the differences are due to inducing seasonal physiologies than the current photoperiod the fish are experiencing. For example, in the abstract the authors write:

Lines 32-35: "We demonstrate in a factorial experiment that long-day photoperiod treatment leads to increased susceptibility to infection and modifies the response to infection for the mucin gene *muc2* and Treg cytokine *foxp3a* in skin tissues."

The wording here implies fish were more susceptible while experiencing long day photoperiods. However, during the infection and data collection, fish in both treatment groups had 12 hrs light and 12 hrs dark - they were not currently experiencing the long-day photoperiod. The authors should take care to not potentially mislead the reader especially throughout the results and discussion.

Minor comments:

1. The authors do a good job at conveying the effect of seasonality on various taxa as requested by Reviewer 2. However, the manuscript would be improved by citing more literature specific to what seasonal cues are known in fish and how it affects their physiology.
2. It would be better to refer to the resistant and susceptible populations as e.g. RES and SUS, rather than the source location of the fish. The location from which the fish were taken is not useful information for the reader and requires translation from CHRU to susceptible and STRU to resistant (this periodically interrupted my train of thought as I was reading the paper, especially in interpreting the results).
3. Line 25: can you be more specific than to just say "seasonal transmission is important"? e.g. "Seasonal transmission is important for x, y, z reasons".
4. Line 72: "Melatonin however appears may be" - this grammatical mistake makes it difficult to interpret what the author is expressing.
5. Line 76-77: how is T3-signalling altered by daylength? What is the effect of long- or short-day length?
6. Line 129: If the fish only live until ~1-2 years old in the wild, it seems they were very old during the experiment. Circadian rhythms are known to dampen in old age (e.g. Manoogian & Panda 2017), however, the authors still show compelling results between groups. It could be worthwhile in the discussion to mention that the results presented here may actually be conservative and could be stronger in younger fish.
7. I think it would be well worth the author's time to make a figure showing the timeline of the experiment. For example, in line 136-137: "Fish were housed individually in 10 litre compartments of 20 litre tanks for a total of 153 days" - where is this total (days) coming from? Is that 153 days starting at the 22-month point (noted on Line 129)? Generally, the timeline is difficult to piece together, and a figure would help the reader understand what happened when throughout the experiment.

Decision letter (RSPB-2020-0288.R0)

02-Mar-2020

Dear Dr Whiting:

I am writing to inform you that we have now obtained responses from referees on manuscript RSPB-2020-0288 entitled "Exposure to long day photoperiods alters immune responses and increases susceptibility to parasitic infection in stickleback" which you submitted to Proceedings B.

Unfortunately, your manuscript has been rejected following full peer review. Your revised manuscript was assessed by two new reviewers because the original reviewers declined the

invitation. One of the new reviewers recommends rejecting the manuscript, the other is critical and recommends major revisions. It is our policy to only allow one round of major revisions and after that we need reviewers to at least be convinced about the findings and the strength of the advance. Of course, new reviewers almost invariably raise new issues when assessing a revised manuscript and it is good practice to allow authors to respond. However, this is only in cases where we otherwise have convergence. Unfortunately, this is not the case here. The reviewers raise a number of new issues, but both highlight (in comments to you and in confidential comments to the Editor) the same crucial methodological point regarding the design of the study that was mentioned as one of the major issues in the previous round. Despite your additional explanation, neither reviewer is convinced. This then leads to too much doubt for us to proceed, hence my decision to reject the manuscript.

Competition for space in Proceedings B is currently extremely severe, as many more manuscripts are submitted to us than we have space to print. We are therefore only able to publish those that are exceptional, convincing and present significant advances of broad interest, and must reject many good manuscripts.

Please find below the comments received from referees concerning your manuscript, not including confidential reports to the Editor. I hope you may find these useful should you wish to submit your manuscript elsewhere.

We are sorry that your manuscript has had an unfavourable outcome, but would like to thank you for offering your work to Proceedings B.

Sincerely,
 Professor Hans Heesterbeek
 mailto: proceedingsb@royalsociety.org

Reviewer(s)' Comments to Author:

Referee: 3

Comments to the Author(s).

This study examines the effect of photoperiod upon immunity and susceptibility to parasitism in three-spined stickleback. Many studies have reported upon seasonal changes in immune function, but very few have employed the use of real parasites and their effect upon host susceptibility. Overall, this study is generally well-presented, although I have some concerns regarding experimental design and presentation of methods that should be addressed.

Major points:

1.) Experimental design- In general, the methods are quite confusing. Can they be simplified? Or at least can an experimental timeline be produced? If I understand the set-up of the experiment correctly, there were two photoperiod treatments, and the LD group were exposed to increasing photoperiods (2 h of light per week) up to 16L:8D, and the 8L:16D groups remained the same. Then, the authors shift photoperiods so that both groups are at 12L:12D, so that groups can be infected by monogeneans experiencing the same photoperiod. The problem with this approach is that SD fish will be photostimulated on 12L:12D and are thus no longer in "winter" condition, and thus are no longer 8L. The LD may still be in reproductive condition, but will be progressing to a non-breeding state because of decreasing photoperiod. Therefore, I argue that this approach confounds assignment of fish to LD or SD. After 20 days of LD exposure (in SD fish), the HPG axis will be fully stimulated. This is corroborated by the SD fish being in more advanced reproductive condition than LD fish in the results. Instead the authors should have probably collected monogeneans from fish that were exposed to 12L:12D, and then transferred to

the 8L and 16L groups. Given the present experimental design, what the authors are really testing is whether photoperiod history has an effect upon host susceptibility. I suppose the paper could be couched in those terms...

2.) Methods- In general, the paper should be re-written so that the methods are more clear. As stated above, an experimental timeline here would be helpful

Minor comments:

Abstract-You never state in the abstract if stickleback are short- or long-day breeders.

Line 73. Delete "appears"

Line 136. Please provide evidence that rapid photoperiod shifts are "stressful" in these fish. Previous studies have rapidly shifted photoperiod, and there seem to be no stress-induced effects.

Line 250-252- And herein lies the confound in this study. How can you term fish SD, when they are in breeding condition?

Line 286-287. Why mention this result if it does not meet the 5% threshold then?

Referee: 4

Comments to the Author(s).

The manuscript by Whiting et al. uses stickleback from susceptible and naïve populations to investigate how long and short-day photoperiods affect parasitic infection dynamics. The authors should be commended for the use of a non-model organism and a naturally occurring parasite. There are some areas in which the author still needs to improve, especially chronobiology wording and its use in presenting results. However, I think these changes should be easy to fix and implement, and don't detract from the novelty of the results. Overall, the work was well executed and presents a valuable contribution to the understanding of the effect of photoperiod on parasitic infection dynamics.

Major comments:

1. Chronobiology terminology is often difficult to accurately use, and I worry some of the usage here may be misleading. For example, the paper argues for differences between LD and SD fish, which is driven by photoperiod manipulations. However, it would be more accurate to claim that the differences are due to inducing seasonal physiologies than the current photoperiod the fish are experiencing. For example, in the abstract the authors write:

Lines 32-35: "We demonstrate in a factorial experiment that long-day photoperiod treatment leads to increased susceptibility to infection and modifies the response to infection for the mucin gene *muc2* and Treg cytokine *foxp3a* in skin tissues."

The wording here implies fish were more susceptible while experiencing long day photoperiods. However, during the infection and data collection, fish in both treatment groups had 12 hrs light and 12 hrs dark - they were not currently experiencing the long-day photoperiod. The authors should take care to not potentially mislead the reader especially throughout the results and discussion.

Minor comments:

1. The authors do a good job at conveying the effect of seasonality on various taxa as

requested by Reviewer 2. However, the manuscript would be improved by citing more literature specific to what seasonal cues are known in fish and how it affects their physiology.

2. It would be better to refer to the resistant and susceptible populations as e.g. RES and SUS, rather than the source location of the fish. The location from which the fish were taken is not useful information for the reader and requires translation from CHRU to susceptible and STRU to resistant (this periodically interrupted my train of thought as I was reading the paper, especially in interpreting the results).

3. Line 25: can you be more specific than to just say “seasonal transmission is important”? e.g. “Seasonal transmission is important for x, y, z reasons”.

4. Line 72: “Melatonin however appears may be” - this grammatical mistake makes it difficult to interpret what the author is expressing.

5. Line 76-77: how is T3-signalling altered by daylength? What is the effect of long- or short-day length?

6. Line 129: If the fish only live until ~1-2 years old in the wild, it seems they were very old during the experiment. Circadian rhythms are known to dampen in old age (e.g. Manoogian & Panda 2017), however, the authors still show compelling results between groups. It could be worthwhile in the discussion to mention that the results presented here may actually be conservative and could be stronger in younger fish.

7. I think it would be well worth the author’s time to make a figure showing the timeline of the experiment. For example, in line 136-137: “Fish were housed individually in 10 litre compartments of 20 litre tanks for a total of 153 days” - where is this total (days) coming from? Is that 153 days starting at the 22-month point (noted on Line 129)? Generally, the timeline is difficult to piece together, and a figure would help the reader understand what happened when throughout the experiment.

Author's Response to Decision Letter for (RSPB-2020-0288.R0)

See Appendix B.

RSPB-2020-1017.R0

Review form: Reviewer 3

Recommendation

Reject - article is scientifically unsound

Scientific importance: Is the manuscript an original and important contribution to its field?

Good

General interest: Is the paper of sufficient general interest?

Good

Quality of the paper: Is the overall quality of the paper suitable?

Marginal

Is the length of the paper justified?

Yes

Should the paper be seen by a specialist statistical reviewer?

No

Do you have any concerns about statistical analyses in this paper? If so, please specify them explicitly in your report.

No

It is a condition of publication that authors make their supporting data, code and materials available - either as supplementary material or hosted in an external repository. Please rate, if applicable, the supporting data on the following criteria.

Is it accessible?

No

Is it clear?

N/A

Is it adequate?

N/A

Do you have any ethical concerns with this paper?

No

Comments to the Author

I appreciate the willingness of authors to consider this study in the context of photoperiod history. However, given the current experimental design of the study, it fails to differentiate whether effects attributed to host immune function are due to prior photoperiod treatments or their current state. In order to tease these effects apart, it would be necessary to have a control group that experienced no prior photoperiod change (i.e. fish that were ramped up to 12L:12D starting March 2016 until the end of infection). Without this control, whether prior photoperiod plays a role is unfortunately confounded by their current state. In other words, are immune responses different because of what fish experienced previously or because of their current physiological state? Unfortunately, the current experiment does not differentiate between these two possibilities.

Review form: Reviewer 4 (Mary L. Westwood)

Recommendation

Accept as is

Scientific importance: Is the manuscript an original and important contribution to its field?

Excellent

General interest: Is the paper of sufficient general interest?

Excellent

Quality of the paper: Is the overall quality of the paper suitable?

Excellent

Is the length of the paper justified?

Yes

Should the paper be seen by a specialist statistical reviewer?

No

Do you have any concerns about statistical analyses in this paper? If so, please specify them explicitly in your report.

No

It is a condition of publication that authors make their supporting data, code and materials available - either as supplementary material or hosted in an external repository. Please rate, if applicable, the supporting data on the following criteria.

Is it accessible?

Yes

Is it clear?

Yes

Is it adequate?

Yes

Do you have any ethical concerns with this paper?

No

Comments to the Author

The authors have implemented all changes from the previous round of revisions. This manuscript provides a thorough and valuable contribution to understanding seasonality and host-parasite interactions. The research is novel and the results are compelling, and will undoubtedly aid future studies in further examining the physiology of seasonality and how it affects host parasite interactions across a diverse array of taxa.

Decision letter (RSPB-2020-1017.R0)

01-Jun-2020

Dear Dr Whiting

I am pleased to inform you that your manuscript RSPB-2020-1017 entitled "Prior exposure to long day photoperiods alters immune responses and increases susceptibility to parasitic infection in stickleback" has been accepted for publication in Proceedings B.

One referee has recommended publication, but the other referee still recommends rejection based on a difference of opinion about the best study design. We have reached a stage where a decision needs to be made because this issue cannot be resolved. As I see it, the scientific debate in your field is helped by publishing the paper, allowing others to respond in the appropriate way. The critical reviewer has reiterated the major concern and I urge you to acknowledge this in your discussion. Therefore, I invite you to respond to the referee's comments and revise your manuscript. Because the schedule for publication is very tight, it is a condition of publication that you submit the revised version of your manuscript within 7 days. If you do not think you will be able to meet this date please let us know.

NB. From April 1 2013, peer reviewed articles based on research funded wholly or partly by RCUK must include, if applicable, a statement on how the underlying research materials – such

as data, samples or models – can be accessed. This statement should be included in the data accessibility section.

[http://datadryad.org/submit?journalID=RSPB&manu=\(Document not available\)](http://datadryad.org/submit?journalID=RSPB&manu=(Document+not+available)) which will take you to your unique entry in the Dryad repository. If you have already submitted your data to dryad you can make any necessary revisions to your dataset by following the above link. Please see <https://royalsociety.org/journals/ethics-policies/data-sharing-mining/> for more details.

Sincerely,
Professor Hans Heesterbeek
mailto:proceedingsb@royalsociety.org

Reviewer(s)' Comments to Author:

Referee: 3

Comments to the Author(s).

I appreciate the willingness of authors to consider this study in the context of photoperiod history. However, given the current experimental design of the study, it fails to differentiate whether effects attributed to host immune function are due to prior photoperiod treatments or their current state. In order to tease these effects apart, it would be necessary to have a control group that experienced no prior photoperiod change (i.e. fish that were ramped up to 12L:12D starting March 2016 until the end of infection). Without this control, whether prior photoperiod plays a role is unfortunately confounded by their current state. In other words, are immune responses different because of what fish experienced previously or because of their current physiological state? Unfortunately, the current experiment does not differentiate between these two possibilities.

Referee: 4

Comments to the Author(s).

The authors have implemented all changes from the previous round of revisions. This manuscript provides a thorough and valuable contribution to understanding seasonality and host-parasite interactions. The research is novel and the results are compelling, and will undoubtedly aid future studies in further examining the physiology of seasonality and how it affects host parasite interactions across a diverse array of taxa.

Author's Response to Decision Letter for (RSPB-2020-1017.R0)

See Appendix C.

Decision letter (RSPB-2020-1017.R1)

08-Jun-2020

Dear Dr Whiting

I am pleased to inform you that your manuscript entitled "Prior exposure to long day photoperiods alters immune responses and increases susceptibility to parasitic infection in stickleback" has been accepted for publication in Proceedings B.

Open Access

You are invited to opt for Open Access, making your freely available to all as soon as it is ready for publication under a CCBY licence. Our article processing charge for Open Access is £1700. Corresponding authors from member institutions (<http://royalsocietypublishing.org/site/librarians/allmembers.xhtml>) receive a 25% discount to these charges. For more information please visit <http://royalsocietypublishing.org/open-access>.

Paper charges

Sincerely,
Proceedings B
<mailto:proceedingsb@royalsociety.org>

Appendix A

Comments to Author:

This manuscript has been evaluated by two expert reviewers, who both agree that the paper presents interesting results. However, both reviewers also point to number of issues that need to be addressed. In particular, both reviewers remark on the currently narrow presentation of the background literature and prevailing hypotheses on seasonality/photoperiod and immune function in vertebrates. In addition, R2 suggests that life history and ecological information about the study species would facilitate data interpretation, while R1 identifies potential flaws in the study design that need further clarification and/or justification. Both reviews also make a number of other suggestions that should be fully considered.

Reviewer(s)' Comments to Author:

Referee: 1

Comments to the Author(s)

The manuscript by Whiting et al. investigated the interactions among photoperiod, immunity and disease susceptibility in a fish species. Specifically, to examine the effects of photoperiod on disease, they lab-reared fish derived from wild populations of three-spined stickleback in either long summer-like photoperiod or short winter-like photoperiods. They then experimentally infected them with the parasite *Gyrodactylus gasterostei* and examined disease clearance, as well as gene expression of inflammatory genes in skin. They showed that long-day housed fish were less capable in clearing parasitic infections and had reduced expression of skin mucin gene (*muc2*) expression and Treg cytokine *fox3a* expression. While there were differences in inflammatory gene expression across populations of fish, photoperiod modulation of infection was consistent across host populations. The authors' suggest these findings demonstrate that photoperiodic modulation of immunity is important for seasonal changes in host susceptibility

Major Issues

One of the major issues, which the authors do admit and address, is whether changes in skin immunity are the cause or consequence of infection. It is quite possible, even likely, that inflammatory responses would be elevated in response to experimental infections. As I said, the author acknowledge this and indicate that their does not does not allow disentangling this relationship; that's said it does weaken my enthusiasm about the putative mechanisms underlying disease resistance in this study.

>>> *It was not our intention in this study to identify mechanisms underlying disease resistance (indeed our design did not allow us to do that), but rather to explore the relationships between day length, measures of immune response and susceptibility to infection, which have never previously been explored in a single experiment. We have done this, and believe our results to be completely novel.*

The design of this study has all individuals being transferred to LD 12:12 following photoperiod manipulation so that they are in a "common photoperiod." While seemingly reasonable on face value, prior photoperiodic conditions drive an animal's response to future day lengths such that a subsequent photoperiod could be perceived as long or short relative to the prior photoperiodic history. 12:12 is particularly troublesome in that it is considered an intermediate photoperiod for many vertebrates and interpreted as long vs short depending on individual. It would have probably been more logical to keep animals in their respective conditions; this would not allow separation of current vs. prior photoperiod, but would avoid the confound of prior photoperiodic history.

>>> *We designed our experiment in this way to avoid confounding the host's response to day length with the parasite's response. There is evidence of seasonal diapause in some parasites*

(e.g. Sommerville & Davey 2002 Can. J. Zool.) and we have noticed that Gyrodactylus appears to breed much more slowly in short day conditions (A.D.C. MacColl, personal observations) and is less able to establish infections. Thus, since in this study we were interested in the responses of the host, we wanted to carry out experimental infections in conditions that were identical for the parasite apart from the physiological state of the host. We have now clarified this in the manuscript at lines 141-145. For the purpose of the experimental demonstration what we have intended to show specifically is that photoperiod-controlled “time of year” physiology has a direct effect on host susceptibility. It therefore doesn’t necessarily matter if the comparison being made is true winter-summer (SD-LD) or late winter-summer (prior SD – prior LD). We have included a statement clarifying that the latter is the most plausible comparison at line 145 and have removed emphasis of the terms “summer” and “winter” from the title and main text.

This last point is more philosophical than procedural. There is a robust literature in vertebrates, especially mammals and birds, looking at seasonal/photoperiodic changes in immune response, and in many of these animal, they show short-day immunoenhancement relative to long days. One of the hypotheses proffered is the winter immunoenhancement hypothesis by which animal have evolved to enhance immunity in short days to buffer against winter stressors. In fact, some causal mechanisms, cortisol, melatonin, adipokines have been suggested. Given this, why did the author decide to couch their findings as long-day suppression rather than short-day enhancement. Admittedly, its arbitrary as to which photoperiod is “baseline” but it struck me as unusual give the previous literature in their field.

>>> As the reviewer states, this is arbitrary. There is also a large and well-established literature that considers suppression of immune responses during reproduction. Because stickleback are long-day breeders and we observed reduced condition in these LD fish, we believe that the framing of long-day susceptibility is concordant. We have now added a brief mention of this issue in the discussion.

Minor Issues

Abstract and intro. While I agree that there are few to no studies looking at photoperiodic changes in susceptibility to infection, despite the ample literature with respect to immune measures, I think its work acknowledging some studies that utilize proxies of infection (e.g., LPC, PolyIC) which are bacterial or viral mimetics, or ex vivo “infection resistance” in the form of ex vivo bacterial killing assays. While these studies do not test actual infection susceptibility with the organism, that have contributed to the seasonal disease literature. Related, I would not agree that seasonality of immune responses generally manifests through changes in innate immunity; there are abundant examples of changes in both humoral and cellular immunity as well.

>>> The abstract and introduction have now been re-written based on the feedback from both reviewers. As a consequence, both concerns raised here are addressed. References related to pathogen mimetics are highlighted at line 81-84.

Abstract, line 29. Change between to among since it’s a comparison of more than two items.

>>> This has been changed.

Given the design of the experiment and the use of F1 animal from susceptible and resistant populations, do all of the experimental animals start out parasite-free?

>>> All animals were raised in the lab from eggs, without contact with infected fish, and thus start out completely naïve to infection, this is highlighted at line 126.

Why are animal re-infected after two days? In other words, if resistance to infection is one of the variables on interest, why is the lack of infection two days later not simply considered as part of this resistance? Is it owing to the time course of putative clearance?

>>> It is standard practice in *Gyrodactylus* infection experiments to re infect after a day or two, to guard against the possibility of stochastic loss of initial infection (De Roij et al 2011). Loss of infection within the first couple of days is more likely a stochastic failure of infection to establish rather than response of the host. There were few fish where this was an issue however.

I understand the desire to call the photoperiods summer photoperiod and winter photoperiod. But because photoperiod and not seasonality was tested here, I suggest using the more standard long-day and short-day as it can be confusing to some that actual seasonality was not assessed in this study.

>>> This has now been changed, although we retain some broad implications of summer and winter in the discussion.

The reproductive response were counterintuitive to me; is this species short-day breeders?

>>> Stickleback are long-day breeders. The pattern we document has arisen because the long-day treatment fish had effectively been through a breeding season during their long-day treatment, while the short-day fish had just begun to respond to the longer days during the infection phase. We have now clarified this and made the distinction clear at lines 97,407,416.

Referee: 2

Comments to the Author(s)

In this manuscript, the authors demonstrated that changes in photoperiod affected parasitic infection dynamics, reproductive physiology, body condition, and immune gene expression in two populations of stickleback that varied in infection susceptibility. This manuscript highlights unique experimental data that will further our understanding of environmental and physiological regulation of immunity and parasite dynamics in a non-model system. Overall, I found the study compelling and applaud the authors' use of a naturally occurring host-parasite system.

General Comments:

1. Throughout the manuscript, I found the scope of the literature cited very narrow, primarily focusing on fish and humans. I would recommend citing more of the mammalian, bird, and even plant literature on seasonality in immune function and disease (e.g., Ashley et al. 2013 (JEB), Fonseca et al. 2014, PLOS ONE).

>>> We have expanded the literature to include examples from more diverse taxa in both the introduction and discussion.

2. I would suggest restructuring the introduction to improve the logical flow of ideas. For example, the first paragraph jumps from talking about fur color to disease to immunity with minimal connections between these ideas. I suggest starting with 1) a broad discussion of seasonal variation in environmental variables (perhaps including seasonality/variation in disease risk and host susceptibility) and then discussing 2) how organisms frequently use photoperiod to time their physiological systems (e.g, reproduction, immunity) to best match the environment to maximize fitness. Finally, since this is an experimental study, I'd recommend including some brief hypotheses and predictions for your study based on the published literature.

>>> We thank the reviewer for this constructive feedback and have incorporated this structuring into the introduction, which has largely been re-written to broaden its focus on literature and hypotheses.

3. I'd recommend incorporating brief life history/ecological information about the stickleback to aid data interpretation. For example, when do wild stickleback normally reproduce? Since your data found that winter sticklebacks are more likely to be sexually mature, this seems like something relevant to discuss if this is an abnormal physiological result. In addition to providing life history on

the stickleback, it would be useful to know a bit more about the parasite itself and its pathogenicity to evaluate the potential selective pressures on the host.

>>> We have provided additional information regarding stickleback life history. In particular, we hope that this clarifies the reproductive condition results within each photoperiod treatment as being consistent with long-day breeding. Further information on parasite pathogenicity can also be found in this section at lines 89-100.

4. For the figure captions, I'd recommend making them a bit more descriptive e.g., "experimental outcomes" is vague.

>>> Figure captions have been simplified and made more descriptive.

5. Given the multitude of immune genes and tissues that were sampled, I'd suggest adding a small table that summarizes the "effect" columns in Tables S3 and S4 to make it easier for the reader to keep track of the major results.

>>> We appreciate this would aid in interpretation, however we would prefer not to include such a table for the following reasons. Firstly, some of the major results cannot be accurately summarised in a simple "Effect" term. For example models with multiple interaction terms. We are also towards the upper end of space restrictions and adding in extra tables will mean losing whole portions of text. Having expanded sections of the introduction and discussion to broaden the literature, the rest of the manuscript has already been condensed substantially. Lastly, we feel the most important model results are already better summarised visually in figures, therefore we feel additional table descriptions would be redundant for the most important model effects.

Specific Comments:

Line 51: "Synchronising temporal variation through photoperiod", do you mean synchronizing physiology?

>>> This has been amended.

Lines 54-55: Since you don't measure or discuss thyroid hormone in the rest of the manuscript, I was distracted by this specific explanation of the mechanism involved.

>>> This has now been removed, but we retain the citation pertaining to a more conserved hormonal mechanism among vertebrates.

Line 58: Please provide a citation for this sentence.

>>> This sentence has been removed.

Line 60: In Line 58, it is mentioned that most of immune seasonality is manifested through changes in innate immunity, but this example in stickleback (adaptive T-cell response), is not within innate immunity. Perhaps rephrase first sentence to reflect this?

>>> This sentence has been removed.

Line 68: "known from controlled studies"—can you provide which species these were done in?

>>> This sentence has been removed.

Lines 154-156: Please provide a sample size for the number of fish that had to be re-infected because previous infection didn't take (particularly since this seemed to require additional handling/manipulation).

>>> Provided at line 178.

Lines 173-175: Please provide a citation that justifies using the measure of reproductive condition (REPRO) in males. Also, how accurate is this measure in determining actual reproductive condition in males?

>>> Broadly, the size and melanisation of testes is a clear indicator of whether males are not at all invested in reproduction (small, pale testes, score = 1) or somewhat invested in reproduction (large, melanised testes, score = 2). The enlargement and melanisation of testes accompanies the external signals of males during the reproductive season (blue eyes, red throat).

The final score of 3, associated with kidney enlargement, corresponds to the final maturation observed in males whereby the kidneys enlarge to produce spiggin for nest-building. This feature is only observed in males who are very close to/have recently bred. We have provided this additional information regarding the biology and include a citation referencing the use of this index in other work at lines 198-201.

Line 192: I found the term “post-reproductive” confusing.

>>> This term has been removed from the sub-heading in favour of something more informative.

Line 324: Since you’re now discussing potential effects of condition and reproductive state, perhaps a new subheading here?

>>> This has been added.

Line 386: If photoperiod drives sexual maturation in these fish, then why are the winter fish more sexually mature than the summer fish? I am also unclear about the mechanism explaining why summer fish are in worse condition than winter fish. Are there other infections/physiological changes that take place in the summer that could explain this decline in condition? Is this an artifact of captivity?

>>> We acknowledge that this is a repeated point of confusion and have added a short explanation as to what we understand to be taking place during photoperiod treatments prior to infection. This is at line 250-252.

Lines 416-425: I acknowledge you can’t determine the exact relationship between muc2 expression and condition given your data, but could you speculate given the literature about why these mucin genes decrease in fish in lower condition? Is this part of the stress response in these fish?

Line 439: Replace “gender-specific” with “sex-specific”

>>> This has been changed.

Lines 473-476: Can you specify some of these other commonly occurring seasonal parasites/diseases? Do any of these affect the fish more severely than *G. gasterostei*?

>>> This information has been added at line 488-489.

Appendix B

02-Mar-2020

Dear Dr Whiting:

I am writing to inform you that we have now obtained responses from referees on manuscript RSPB-2020-0288 entitled "Exposure to long day photoperiods alters immune responses and increases susceptibility to parasitic infection in stickleback" which you submitted to Proceedings B.

Unfortunately, your manuscript has been rejected following full peer review. Your revised manuscript was assessed by two new reviewers because the original reviewers declined the invitation. One of the new reviewers recommends rejecting the manuscript, the other is critical and recommends major revisions. It is our policy to only allow one round of major revisions and after that we need reviewers to at least be convinced about the findings and the strength of the advance. Of course, new reviewers almost invariably raise new issues when assessing a revised manuscript and it is good practice to allow authors to respond. However, this is only in cases where we otherwise have convergence. Unfortunately, this is not the case here. The reviewers raise a number of new issues, but both highlight (in comments to you and in confidential comments to the Editor) the same crucial methodological point regarding the design of the study that was mentioned as one of the major issues in the previous round. Despite your additional explanation, neither reviewer is convinced. This then leads to too much doubt for us to proceed, hence my decision to reject the manuscript.

Competition for space in Proceedings B is currently extremely severe, as many more manuscripts are submitted to us than we have space to print. We are therefore only able to publish those that are exceptional, convincing and present significant advances of broad interest, and must reject many good manuscripts.

Please find below the comments received from referees concerning your manuscript, not including confidential reports to the Editor. I hope you may find these useful should you wish to submit your manuscript elsewhere.

We are sorry that your manuscript has had an unfavourable outcome, but would like to thank you for offering your work to Proceedings B.

Sincerely,
Professor Hans Heesterbeek
mailto: proceedingsb@royalsociety.org

Reviewer(s)' Comments to Author:

Referee: 3

Comments to the Author(s).

This study examines the effect of photoperiod upon immunity and susceptibility to parasitism in three-spined stickleback. Many studies have reported upon seasonal

changes in immune function, but very few have employed the use of real parasites and their effect upon host susceptibility. Overall, this study is generally well-presented, although I have some concerns regarding experimental design and presentation of methods that should be addressed.

Major points:

1.) Experimental design- In general, the methods are quite confusing. Can they be simplified? Or at least can an experimental timeline be produced? If I understand the set-up of the experiment correctly, there were two photoperiod treatments, and the LD group were exposed to increasing photoperiods (2 h of light per week) up to 16L:8D, and the 8L:16D groups remained the same. Then, the authors shift photoperiods so that both groups are at 12L:12D, so that groups can be infected by monogeneans experiencing the same photoperiod. The problem with this approach is that SD fish will be photostimulated on 12L:12D and are thus no longer in "winter" condition, and thus are no longer 8L. The LD may still be in reproductive condition, but will be progressing to a non-breeding state because of decreasing photoperiod. Therefore, I argue that this approach confounds assignment of fish to LD or SD. After 20 days of LD exposure (in SD fish), the HPG axis will be fully stimulated. This is corroborated by the SD fish being in more advanced reproductive condition than LD fish in the results. Instead the authors should have probably collected monogeneans from fish that were exposed to 12L:12D, and then transferred to the 8L and 16L groups. Given the present experimental design, what the authors are really testing is whether photoperiod history has an effect upon host susceptibility. I suppose the paper could be couched in those terms...

>>> We have endeavoured to simplify the methods, and have included as suggested an experimental timeline that outlines the experiment (Figure 1, which was previously S1). It is clear that there are at least two ways in which we could have done this experiment: either as we did, with the infection experiment carried out in common day length conditions or, as the reviewer suggests, with the fish maintained in their day length treatments and the parasite exposed to different day lengths. We believe that both would have been completely novel, and that neither design is right or wrong, although the inference may be different. In our design we at least know that the effect we document had to have been mediated through the fish host, which would not have been true for the alternative. Because we were interested in the effect of photoperiod on hosts specifically, the experimental design had to reflect this. In addition, as the reviewer points out, there is some semantic difficulty in how to describe the short-day and long-day treatments (photoperiod versus photoperiod history). We have therefore dropped the short/long day (SD/LD) terminology and replaced it by 'prior-short day' (PSD) and 'prior-long day' (PLD).

2.) Methods- In general, the paper should be re-written so that the methods are more clear. As stated above, an experimental timeline here would be helpful

>>> The timeline has now been added as Figure 1 in response to both reviewers.

Minor comments:

Abstract-You never state in the abstract if stickleback are short- or long-day breeders.

>>> Now added.

Line 73. Delete “appears”

>>> Amended.

Line 136. Please provide evidence that rapid photoperiod shifts are “stressful” in these fish. Previous studies have rapidly shifted photoperiod, and there seem to be no stress-induced effects.

>>> We have removed the mention that rapid shifts are stressful.

Line 250-252- And herein lies the confound in this study. How can you term fish SD, when they are in breeding condition?

>>> Again, this is a semantic issue rather than a confounding factor. We term the fish SD (short-day) as these fish were exposed to short days instead of long days prior to infection. We have now amended the text to remove references to SD and LD in order to prevent this confusion.

Line 286-287. Why mention this result if it does not meet the 5% threshold then?

>>> We believe it is still of interest to the reader to report how many infections were cleared alongside final parasite burdens, and whether clearing infection varied by treatment. Final models (selected based on AIC) included photoperiod treatment as an explanatory variable, therefore we feel it is necessary to report the difference in infection clearing between photoperiod treatment groups. Further, in light of the accepted 5% threshold being statistically arbitrary, we see no reason not to report this result alongside the caveat that it very narrowly misses out on meeting this arbitrary threshold.

Referee: 4

Comments to the Author(s).

The manuscript by Whiting et al. uses stickleback from susceptible and naïve populations to investigate how long and short-day photoperiods affect parasitic infection dynamics. The authors should be commended for the use of a non-model organism and a naturally occurring parasite. There are some areas in which the author still needs to improve, especially chronobiology wording and its use in presenting results. However, I think these changes should be easy to fix and implement, and don't detract from the novelty of the results. Overall, the work was well executed and presents a valuable contribution to the understanding of the effect of photoperiod on parasitic infection dynamics.

Major comments:

1. Chronobiology terminology is often difficult to accurately use, and I worry some of the usage here may be misleading. For example, the paper argues for differences between LD and SD fish, which is driven by photoperiod manipulations. However, it would be more accurate to claim that the differences are due to inducing seasonal physiologies than the current photoperiod the fish are experiencing. For example, in the abstract the authors write:

Lines 32-35: “We demonstrate in a factorial experiment that long-day photoperiod treatment leads to increased susceptibility to infection and modifies the response to infection for the mucin gene *muc2* and Treg cytokine *foxp3a* in skin tissues.”

>>> We appreciate the potential confusion surrounding some of the terminology used and have endeavoured to adjust this throughout the manuscript. We have removed any mention of LD and SD and instead refer to treatments as PLD and PSD (see above).

The wording here implies fish were more susceptible while experiencing long day photoperiods. However, during the infection and data collection, fish in both treatment groups had 12 hrs light and 12 hrs dark - they were not currently experiencing the long-day photoperiod. The authors should take care to not potentially mislead the reader especially throughout the results and discussion.

>>> Here we have amended the abstract to read “previous exposure to long-day photoperiod treatment”, and have taken a similar approach throughout.

Minor comments:

1. The authors do a good job at conveying the effect of seasonality on various taxa as requested by Reviewer 2. However, the manuscript would be improved by citing more literature specific to what seasonal cues are known in fish and how it affects their physiology.

>>> We have added additional information and references regarding seasonal cues and physiology in fish at lines 76-78.

2. It would be better to refer to the resistant and susceptible populations as e.g. RES and SUS, rather than the source location of the fish. The location from which the fish were taken is not useful information for the reader and requires translation from CHRU to susceptible and STRU to resistant (this periodically interrupted my train of thought as I was reading the paper, especially in interpreting the results).

>>> We agree this aids readability and have adjusted as suggested.

3. Line 25: can you be more specific than to just say “seasonal transmission is important”? e.g. “Seasonal transmission is important for x, y, z reasons”.

>>> Due to restricted space in the abstract following the addition of further information regarding experimental design and photoperiod treatment, we have chosen to remove this mention from the abstract. We believe this is a relatively minor point to make in the abstract and is expanded on anyway in the first paragraph of the introduction.

4. Line 72: “Melatonin however appears may be” – this grammatical mistake makes it difficult to interpret what the author is expressing.

>>> Amended

5. Line 76-77: how is T3-signalling altered by daylength? What is the effect of long- or short-day length?

>>> Mention of this has now been removed as we believe it is likely an over-specific example of immune modulation by photoperiod that is not relevant here. Instead,

we cite a review that highlights several ways in which immune responses may be modulated by photoperiod.

6. Line 129: If the fish only live until ~1-2 years old in the wild, it seems they were very old during the experiment. Circadian rhythms are known to dampen in old age (e.g. Manoogian & Panda 2017), however, the authors still show compelling results between groups. It could be worthwhile in the discussion to mention that the results presented here may actually be conservative and could be stronger in younger fish.

>>> We thank the reviewer for this insight and have included a short mention of this in the discussion.

7. I think it would be well worth the author's time to make a figure showing the timeline of the experiment. For example, in line 136-137: "Fish were housed individually in 10 litre compartments of 20 litre tanks for a total of 153 days" - where is this total (days) coming from? Is that 153 days starting at the 22-month point (noted on Line 129)? Generally, the timeline is difficult to piece together, and a figure would help the reader understand what happened when throughout the experiment.

>>> This figure did already exist in part as Figure S1, however based on comments from both reviewers we have updated it with additional information and moved it into the main document as Figure 1. We now also reference this figure within the methods to aid interpretation of the timeline.

Journal Name: Proceedings of the Royal Society B

Journal Code: RSPB

Print ISSN: 0962-8452

Online ISSN: 1471-2954

MS Reference Number: RSPB-2020-0288

Article Status: REJECTED

MS Dryad ID: RSPB-2020-0288

MS Title: Exposure to long day photoperiods alters immune responses and increases susceptibility to parasitic infection in stickleback

MS Authors: Whiting, James; Mahmud, Muayad; Bradley, Janette; MacColl, Andrew

Contact Author: James Whiting

Contact Author Email: j.whiting2@exeter.ac.uk

Contact Author Address 1: Geoffrey Pope Building

Contact Author Address 2: University of Exeter

Contact Author Address 3:

Contact Author City: Exeter

Contact Author State:

Contact Author Country: United Kingdom of Great Britain and Northern Ireland

Contact Author ZIP/Postal Code: EX4 4QJ

Keywords: seasonal infection, photoperiod, seasonal immune responses, stickleback, host-parasite interactions

Abstract: Seasonal disease and parasitic infection are common phenomena across many organisms, including humans. Seasonal transmission is important and there is increasing evidence for intrinsic seasonal variation in immune systems. Changes are orchestrated through organisms' physiological clocks using cues such as day length, and

ample research has demonstrated multiple immune responses are modulated by photoperiod in diverse taxa. But to date, there have been few experimental demonstrations that seasonal cues alter susceptibility to infection. We investigated the interactions among photoperiod, immunity, and susceptibility in three-spined stickleback, an aquatic vertebrate model reared in laboratory conditions, and its external, directly-reproducing, monogenean parasite *Gyrodactylus gasterostei*. We demonstrate in a factorial experiment that long-day photoperiod treatment leads to increased susceptibility to infection and modifies the response to infection for the mucin gene *muc2* and Treg cytokine *foxp3a* in skin tissues. Expression of skin *muc2* is reduced in long-day fish, and negatively associated with parasite abundance in infected individuals. We also observe inflammatory gene expression variation associated with natural inter-population variation in resistance, but find that the photoperiod modulation of susceptibility is consistent across host populations. Thus, photoperiod-modulation of the response to infection is important for host susceptibility, highlighting new mechanisms affecting seasonality of host-parasite interactions.

EndDryadContent

Appendix C

01-Jun-2020

Dear Dr Whiting

I am pleased to inform you that your manuscript RSPB-2020-1017 entitled "Prior exposure to long day photoperiods alters immune responses and increases susceptibility to parasitic infection in stickleback" has been accepted for publication in Proceedings B.

One referee has recommended publication, but the other referee still recommends rejection based on a difference of opinion about the best study design. We have reached a stage where a decision needs to be made because this issue cannot be resolved. As I see it, the scientific debate in your field is helped by publishing the paper, allowing others to respond in the appropriate way. The critical reviewer has reiterated the major concern and I urge you to acknowledge this in your discussion. Therefore, I invite you to respond to the referee's comments and revise your manuscript. Because the schedule for publication is very tight, it is a condition of publication that you submit the revised version of your manuscript within 7 days. If you do not think you will be able to meet this date please let us know.

- 1) A text file of the manuscript (doc, txt, rtf or tex), including the references, tables (including captions) and figure captions. Please remove any tracked changes from the text before submission. PDF files are not an accepted format for the "Main Document".

- 2) A separate electronic file of each figure (tiff, EPS or print-quality PDF

preferred). The format should be produced directly from original creation package, or original software format. PowerPoint files are not accepted.

3) Electronic supplementary material: this should be contained in a separate file and where possible, all ESM should be combined into a single file. All supplementary materials accompanying an accepted article will be treated as in their final form. They will be published alongside the paper on the journal website and posted on the online figshare repository. Files on figshare will be made available approximately one week before the accompanying article so that the supplementary material can be attributed a unique DOI.

If you wish to submit your data to Dryad (<http://datadryad.org/>) and have not already done so you can submit your data via this

link [http://datadryad.org/submit?journalID=RSPB&manu=\(Document not available\)](http://datadryad.org/submit?journalID=RSPB&manu=(Document not available)) which will take you to your unique entry in the Dryad repository. If you have already submitted your data to dryad you can make any necessary revisions to your dataset by following the above link.

Please see <https://royalsociety.org/journals/ethics-policies/data-sharing-mining/> for more details.

6) For more information on our Licence to Publish, Open Access, Cover images and Media summaries, please

visit <https://royalsociety.org/journals/authors/author-guidelines/>.

Sincerely,

Professor Hans Heesterbeek

Reviewer(s)' Comments to Author:

Referee: 3

Comments to the Author(s).

I appreciate the willingness of authors to consider this study in the context of photoperiod history. However, given the current experimental design of the study, it fails to differentiate whether effects attributed to host immune function are due to prior photoperiod treatments or their current state. In order to tease these effects apart, it would be necessary to have a control group that experienced no prior photoperiod change (i.e. fish that were ramped up to 12L:12D starting March 2016 until the end of infection). Without this control, whether prior photoperiod plays a role is unfortunately confounded by their current state. In other words, are immune responses different because of what fish experienced previously or because of their current physiological state? Unfortunately, the current experiment does not differentiate between these two possibilities.

>>> We thank the reviewer for their comments in the previous round which we believe have better framed our results and manuscript on the whole. We accept that there will likely be discussion over aspects of the experimental design given the novelty of this experiment and its findings. We disagree that no distinction can be made between current physiological state (12L:12D) and prior photoperiod (16L:8D or 8L:16D). If the former were the prevailing effect, there would no expectation for the observed

susceptibility differences between photoperiod treatment groups because all individuals shared the same 12L:12D treatment for infection. To say otherwise is to ignore that, within source population groups, prior photoperiod history was the sole difference between fish that were statistically more and less susceptible to infection. The reviewer wishes for there to be a clear distinction between “what fish experienced previously” and “their current physiological state”, but such a distinction is unnecessary. As we make clear throughout, the current physiological states of our fish, which are clearly different, are the product of their previous photoperiod history. In nature, that an individual’s physiology is a product of its photoperiod history (the lengthening or shortening of days) is a fundamental facet of circannual rhythm and seasonality. The experimental design makes clear that photoperiod history is adequate to induce these physiological states, which experience differences in parasite susceptibility, when all else is equal.

Further, it is unclear what the expected result for such a control treatment would be, and how this would aid interpretation of results. Would the results from the control treatment be more similar to those from our prior long day, our prior short day, or somewhere intermediate? And what would this tell us about the role of photoperiod on infection susceptibility that the current results do not?

However, in light of these comments we agree that it is necessary to express the caveats of the experimental design again in our discussion, where we also suggest future work that may further address questions on this topic. This can now be found at line 459.

Referee: 4

Comments to the Author(s).

The authors have implemented all changes from the previous round of revisions. This manuscript provides a thorough and valuable contribution to understanding seasonality and host-parasite interactions. The research is novel and the results are compelling, and will undoubtedly aid future studies in further examining the physiology of seasonality and how it affects host parasite interactions across a diverse array of taxa.

>>> We’d like to thank the reviewer again for their feedback in the previous round of comments and help in improving this manuscript.